# Is the acquisition worth the cost? Surrogate losses for Consistent Two-stage Classifiers

**Florence Regol**
Block, Toronto, Canada and McGill University, ILLS, Mila
`florence.robert-regol@mail.mcgill.ca`

**Joseph Cotnareanu**
McGill University, ILLS, Mila

**Theodore Glavas**
McGill University, ILLS, Mila

**Mark Coates**
McGill University, ILLS, Mila

## Abstract

Recent years have witnessed the emergence of a spectrum of foundation models, covering a broad range of capabilities and costs. Often, we effectively use foundation models as feature generators and train classifiers that use the outputs of these models to make decisions. In this paper, we consider an increasingly relevant setting where we have two classifier stages. The first stage has access to features $x$ and has the option to make a classification decision or defer, while incurring a cost, to a second classifier that has access to features $x$ and $z$. This is similar to the "learning to defer" setting, with the important difference that we train both classifiers jointly, and the second classifier has access to more information. The natural loss for this setting is an $\ell_{01c}$ loss, where a penalty is paid for incorrect classification, as in $\ell_{01}$, but an additional penalty $c$ is paid for consulting the second classifier. The $\ell_{01c}$ loss is unwieldy for training. Our primary contribution in this paper is the derivation of a hinge-based surrogate loss $\ell_{hinge}^c$ that is much more amenable to training but also satisfies the property that $\ell_{hinge}^c$-consistency implies $\ell_{01c}$-consistency.

## 1 Introduction

With the emergence of a spectrum of foundation models, covering a broad range of capabilities and costs, we are increasingly faced with a decision as to which model to use. For example, can we make a decision locally, on an edge device, or should we incur the additional communication and computational cost of sending the query to a more powerful remote model? In many cases, we use the pre-trained foundation model essentially as a feature generator, and strive to train a classifier that uses the output of the foundation model as its input. In this setting, we then face a task of training two classifiers, while simultaneously learning when to defer to the more powerful model.

One approach to solve this problem is to 1) train the more powerful classifier first; and 2) train the decision module with the smaller classifier afterwards (either jointly or separately). This strategy has proven successful and benefits from strong theoretical foundations [Keswani et al., 2021, Wilder et al., 2021, Verma et al., 2022, Mao et al., 2023, 2024b], but intuitively, this appears inefficient. Indeed, with this approach, both classifiers expend effort exploring regions of the input space where their predictions will ultimately not be used. Because of this, it is important to consider and formalize the problem where the classifiers and the module deciding which model to use are trained jointly.

While there has been a significant body of work establishing consistent losses for the related problem of classification with learning to defer to experts or with reject [Verma et al., 2022, Herbei and Wegkamp, 2006], these losses do not cover the case where we jointly train multiple inference

39th Conference on Neural Information Processing Systems (NeurIPS 2025).

classifiers along with the decision module. In these well-studied settings, the task is to learn one classifier and to either defer to an oracle (learning to reject) [Chow, 1970], to an expert (learning to defer) [Madras et al., 2018] or to multiple experts (learning to defer to multiple experts) [Verma et al., 2022]. However, these frameworks assume that the experts are external to the problem setting. They do not address how to train the experts alongside the base classifier.

In this work, we address the problem of jointly training two classifiers with a decision module. The two classifiers incur different costs, with the implication that the more expensive classifier offers better performance. We model this problem by introducing an additional information variable, $Z$, which represents the extra information available to the more powerful classifier. The decision module and the base classifier both have access to the same base input variable $X$. We refer to this setup as the *two-stage classification* problem.

We provide the optimal solution to this problem, as well as a surrogate loss function that is more suitable for training. The surrogate loss, which is based on the hinge loss, aligns with the standard cost-aware $0 - 1$ loss formulation commonly used in classification tasks. We validate our theoretical findings on synthetic datasets and demonstrate the practical relevance of the problem by presenting results on a standard large language model (LLM) task, where two LLMs of varying sizes are used to answer multi-question math problems. Additionally, we provide a proof that the cross-entropy loss, which is sometimes used heuristically in existing literature, is not Bayes consistent with the natural $0 - 1c$ loss, further justifying the need to explore this problem at a theoretical level.

Our main contributions are as follows:

1. We formulate a problem setting for learning a model that integrates two classifiers, where one has access to additional information but comes with a cost $c$. The goal is to train the models and simultaneously learn the decision function to determine whether to consult the more powerful classifier for a given sample.

2. We present a surrogate loss function based on the hinge loss, which is suitable for training with cost-aware classification tasks. We show that it is consistent with respect to the $0 - 1c$ loss that is natural for the considered problem.

3. We validate the theoretical findings, which are the primary contribution of the work, on synthetic datasets and provide practical insights through experiments on a standard LLM task.

## 2 Related Work

Loss consistency is an important topic that has been widely explored, as it serves as the fundamental link between the loss we optimize in practice and the actual loss we aim to minimize. Foundational results have been established for classical risks [Steinwart, 2007, Tewari and Bartlett, 2007, Bartlett et al., 2006], and the emergence of new target losses has prompted the development of new consistency results. Learning to defer (L2D) is a wide category of settings in which the task is to learn a classifier and a deferral rule, either to reject (learning to abstain) [Chow, 1957, 1970, Herbei and Wegkamp, 2006, Cao et al., 2022, Wiener and El-Yaniv, 2011, Geifman and El-Yaniv, 2019] or to defer to one or more experts of varying costs [Madras et al., 2018, Keswani et al., 2021, Wilder et al., 2021].

Mozannar and Sontag [2020] were the first to provide Bayes-consistency results for their proposed generalized cross entropy loss for learning to defer, followed by Verma et al. [2022], who used a one-vs-all loss. Awasthi et al. [2022] explored stronger guarantees than Bayes-consistency by introducing $\mathcal{H}$-consistency bounds [Long and Servedio, 2013]. Mozannar et al. [2023] prove that earlier approaches, such as [Mozannar and Sontag, 2020, Verma et al., 2022], fall short of realizable $\mathcal{H}$-consistency, and propose a new algorithm without a Bayes-consistency proof. This shortcoming is addressed by Mao et al. [2024a], who recently published a unifying work, introducing a new family of surrogate losses for the learning to defer problem with a single expert, and providing Bayes-consistency, realizable $\mathcal{H}$-consistency, and $\mathcal{H}$-consistency bounds. Verma et al. [2022] extended the work of Mozannar and Sontag [2020], Verma et al. [2022] to the multi-expert setting with Bayes-consistency guarantees. Mao et al. [2024b] introduced general cost functions and surrogate losses, extending the results of Mozannar and Sontag [2020] with $\mathcal{H}$-consistency bounds for joint training, and offering stronger guarantees than Bayes-consistency. Mao et al. [2023] also provided $\mathcal{H}$-consistency bounds for a slightly different setting where training the classifier and the deferral rule is done separately.

It appears that consistency results have been thoroughly studied in the case where only a single classifier is trained. However, the setting where multiple classifiers are trained jointly with a decision module has been largely neglected. This type of architecture is used in adaptive computation or dynamic networks [Han et al., 2022a], a branch of research focused on developing architectures that adaptively allocate computation. Since the main objective is to improve average inference efficiency, such dynamic network architectures have attracted significant interest in the development of scalable LLM inference [Liu et al., 2023, Elbayad et al., 2020, Zeng et al., 2024, Xia et al., 2024, Leviathan et al., 2023, Chen et al., 2024].

Although there is growing interest in these types of networks, the losses used to train such models are mostly heuristic and lack strong theoretical foundations. In practice, these models typically train the classifiers separately and rely on threshold-based decisions [Han et al., 2022a, Schuster et al., 2022]. Some theoretical research has been conducted on this separated training approach: Jitkrittum et al. [2023] explored the connection between threshold decisions and risk under a principled 0-1 loss, identifying conditions under which the two coincide. However, this separate training is not guaranteed to be the best approach. In fact, the importance of jointly learning the classifiers and the decision module has been empirically demonstrated [Han et al., 2022b, Yu et al., 2022, Regol et al., 2024, Krzepkowski et al., 2024], motivating the development of joint learning approaches [Regol et al., 2024] and classifier-deferral-aware training methods [Han et al., 2022b, Yu et al., 2022]. These works lack a connection to surrogate losses and a well-defined risk framework, which is the gap we aim to address in this work.

## 3  The two-stage classification problem

We consider a setting of two-stage classification where there are two classifiers: $f_1$ and $f_2$. The second classifier, $f_2$, has access to additional information $z$, but it also incurs an additional cost, denoted as $c$. We therefore have the choice between using the prediction of the first classifier, or to pay the additional cost and then use the more informed second classifier.

In practice, $z$ can be explicitly modeled as an additional input signal or feature, which may come with higher access costs. For instance, in recommendation systems, different types of user data queries can vary significantly in terms of latency and infrastructure expense. A common approach is to first run a lightweight model for initial inference, and then selectively identify instances that would benefit from a more complex model with access to richer features. This tiered architecture is notably used by Youtube's recommendation system [Covington et al., 2016], for instance.

Alternatively, $z$ can conceptually represent the augmented modeling capability of a larger model that has more parameters and/or was trained on a larger data set.

Denote by $\mathcal{X}$ the feature space, $\mathcal{Z}$ the additional information space, and $\mathcal{Y} = \{1, \ldots, K\}$ the label space. We are given instance-label-information triples $\{(x_i, z_i, y_i)\}_{i=1}^n$ independently and identically drawn from an underlying distribution $\mathcal{D}$ with probability function $p(X, Z, Y)$. We additionally introduce the decision module $r : \mathcal{X} \to 0, 1$, which indicates whether we are using, for the final decision, the first classifier $f_1$ if $r(x) = 0$ or to defer to the second classifier $f_2$ if $r(x) = 1$.

The goal of two-stage classification is to train a two-stage classifier $f : \mathcal{X} \times \mathcal{Z} \to \mathcal{Y}$ that encompasses both the classifiers $f_1 : \mathcal{X} \to \mathcal{Y}$, $f_2 : \mathcal{X} \times \mathcal{Z} \to \mathcal{Y}$, and the decision module $r$. The set $\mathcal{H}$ of two-stage classifiers is therefore defined as follows:

$$\mathcal{H} = \{f : f(x, z) = \begin{cases} f_1(x), & r(x) = 0 \\ f_2(x, z), & r(x) = 1. \end{cases} \tag{1}$$

The loss associated with such a setting is the zero-one-exit loss $\ell_{01c}$, which can be expressed as a variant of the traditional zero-one loss $\ell_{01}(f(\cdot), y) = \mathbb{1}[f(\cdot) \neq y]$:

$$\ell_{01c}(f(x, z), y) = \begin{cases} \mathbb{1}[f_1(x) \neq y], & r(x) = 0, \\ \mathbb{1}[f_2(x, z) \neq y] + c, & r(x) = 1, \end{cases} \tag{2}$$

where $\mathbb{1}[\cdot]$ is the indicator function. The cost $c$ can be an instance-specific function, i.e., $c(x)$, provided it is known and deterministic. Since the additional information $z$ is only accessible at a cost $c$, the first classifier and the decision function do not have access to it; the classifiers $f_1(x)$ and $r(x)$ take only $x$ as input.

Our task is to train a two-stage classifier $f \in \mathcal{H}$, as defined by (1), that can minimize the expectation of $\ell_{01c}$ over the data distribution. The risk is:

$$R_{01c}(f) = \mathbb{E}_{p(x,z,y)}[\ell_{01c}(f(x,z), y)], \tag{3}$$

and its optimal value $R_{01c}^* = R_{01c}(f^*)$ is obtained by the Bayes-optimal classifier:

$$f^* = \arg\min_{f \in \mathcal{H}} R_{01c}(f). \tag{4}$$

The $01c$ loss is discrete, and thus difficult to work with. We would like to be able to identify a surrogate loss $\ell_\phi$ such that $\ell_\phi$-consistency implies $\ell_{01c}$-consistency. This is our main contribution in this work. We specify a surrogate loss function that satisfies this property, and show that other heuristic surrogate losses that are used in the literature for joint training [Regol et al., 2024, Ding et al., 2024] do not. Taking a step beyond this, we specify how to construct and train a two-stage classifier using the posited surrogate loss and present empirical results to validate our result.

### 3.1 The solution

We start by providing the solution to the optimization problem specified by (4). We first define a compact notation for the posteriors:

$$\eta_y(x) \triangleq p(Y = y|x), \qquad \zeta_y(x,z) \triangleq p(Y = y|x,z). \tag{5}$$

**Lemma 3.1.** *The optimal solution $f^* = \arg\min_{f \in \mathcal{H}} R_{01c}(f)$ is the following:*

$$f^* = \begin{cases} \arg\max_y \eta_y(x), & \text{if } \max_y \eta_y(x) \geq \mathbb{E}_{p(Z|x)}[\max_y \zeta_y(x,Z)] - c, \\ \arg\max_y \zeta_y(x,z), & \text{else}. \end{cases} \tag{6}$$

*See Appendix A.4 for the proof of the lemma.*

Using our previous definition of a two-stage classifier, this would correspond to:

$$f_1^*(x) = \arg\max_y \eta_y(x) \text{ if } \{x; r^*(x) = 0\} \tag{7}$$

$$f_2^*(x,z) = \arg\max_y \zeta_y(x,z) \text{ if } \{x; r^*(x) = 1\} \tag{8}$$

$$r^*(x) = \begin{cases} 0 & \text{if } \max_y \eta_y(x) \geq \mathbb{E}_{p(Z|x)}[\max_y \zeta_y(x,Z)] - c, \\ 1 & \text{o.w.} \end{cases} \tag{9}$$

The optimal solution is interesting. It hints towards a model that is slightly different from most existing methods. Yes, the optimal decision should depend on $p_{\max} = \max \eta(x)$, but the threshold for $p_{\max}$ should be set based on the *expected future gain*: $\tau < \mathbb{E}_{p(Z|x)}[\max \zeta(x,Z)] - c$. In that setting, $r(x)$ should identify the set on which $\max \eta(x) \geq \mathbb{E}_{p(z|x)} \max \zeta(x,z) - c$, and, for these elements only, it should select the class with the largest probability according to the posterior $\eta(x)$. This result is similar to the solution for the decision rule given fixed classifiers first provided by Jitkrittum et al. [2023], which would read as $\max_y \eta_y(x) \geq \max_y \zeta_y(x,Z) - c$. However, our explicit modeling of the two-tiered information available to $f_1, r$ and $f_2$ provides a more practical and detailed solution, as it allows us to integrate the constraint that $r$ cannot fully access the information available to $f_2$. This modeling choice leads to a decision based on the **expected** future gain.

Unsurprisingly, the first and second classifiers simply predict the class with the highest probability according to their respective posteriors, but only for the samples assigned to them.

## 4 The proposed hinge-based surrogate loss

A common strategy to develop a consistent loss for more complex risk functions is to propose a surrogate loss and verify its consistency. This strategy was employed by early work for the learning to defer problem [Mozannar and Sontag, 2020, Verma and Nalisnick, 2022].

Our proposed surrogate loss is built on a multiclass version of the hinge loss [Tarigan and van de Geer, 2008]. We chose this version because it is Bayes-consistent, unlike other multiclass hinge

losses. We use a hinge loss rather than the more popular cross entropy is because of its linear scaling, which allows to account for the cost in an additive way as in Eqn. 2. Following the definition of the multiclass hinge loss from [Tarigan and van de Geer, 2008], the classifiers are based on $K$-dimensional real valued outputs $\mathbf{t}(x), \mathbf{v}(x,z) \in \mathbb{R}^K$, with the constraints that $\sum_{i=1}^K \mathbf{t}_i(x) = 0$, and $\sum_{i=1}^K \mathbf{v}_i(x,z) = 0$. The label prediction is obtained by returning the max element of the vector. The decision function $\tilde{r}(x)$ returns a real value bounded between 0 and 1. For brevity, we omit the dependence on the inputs and only write $\mathbf{t}, \mathbf{v}$. We can therefore introduce the link function $\varphi$ that connects the real valued output and a soft decision function $\tilde{r}(x)$ to a two-stage classifier function :

$$f = \{f_1, f_2, r\} = \varphi(\mathbf{t}, \mathbf{v}, \tilde{r}) = \{\max_{y \in \mathcal{Y}} \mathbf{t}_y(x), \max_{y \in \mathcal{Y}} \mathbf{v}_y(x,z), \mathbb{1}[\tilde{r}(x) \geq 0.5]\}. \tag{10}$$

Letting $[x]_+ = \max(x, 0)$, our proposed hinge-based surrogate loss is given by:

$$\ell_{hinge}^c(\mathbf{t}, \mathbf{v}, \tilde{r}, x, z, y) = (1 - \tilde{r}(x)) \sum_{y' \neq y} [\mathbf{t}_{y'} + \frac{1}{K-1}]_+ + \tilde{r}(x) \left( \sum_{y' \neq y} [\mathbf{v}_{y'} + \frac{1}{K-1}]_+ + \frac{Kc}{K-1} \right). \tag{11}$$

The loss is composed of a sum of two terms: the first trains the first classifier, and the second trains the second classifier. The balance or weight assigned to each term on a per-sample basis is intuitively controlled by the learned soft decision $\tilde{r}$. If $\tilde{r}(x)$ indicates that a sample should be inferred by $f_1$, then $f_1$ will receive more weight during training at that point. In the second term, corresponding to $f_2$, we include an additional fixed term $\frac{Kc}{K-1}$ that encodes the penalty of using the second classifier. This encourages $\tilde{r}(\cdot)$ to favor the first term unless the benefit of using $f_2$ outweighs the cost. We can then define the associated risk as:

$$R_{hinge}(\mathbf{t}, \mathbf{v}, \tilde{r}) = \mathbb{E}_{p(x,z,y)}[\ell_{hinge}^c(\mathbf{t}, \mathbf{v}, \tilde{r}, x, z, y)], \tag{12}$$

and consider the triplet of minimizers $\mathbf{t}^*(x), \mathbf{v}^*(x,z), \tilde{r}^*(x)$ of such a risk:

$$\mathbf{t}^*, \mathbf{v}^*, r^* = \underset{\mathbf{t}, \mathbf{v} \in \mathbb{R}^K, r \in [0,1]}{\arg\min} R_{hinge}(\mathbf{t}, \mathbf{v}, \tilde{r}), \tag{13}$$

$$f_{hinge}^* = \varphi(\mathbf{t}^*, \mathbf{v}^*, r^*). \tag{14}$$

In the following theorem, we establish the consistency of our proposed surrogate loss w.r.t. $\ell_{01c}$, meaning that if a learned two-stage classifier $f$ converges to the optimal surrogate risk $R_{hinge}^*$, it also converges to the optimal target risk $R_{01c}^*$.

**Theorem 4.1.** *There exists a link function $\varphi$ s.t. for any distribution $p(x,z,y)$, we have that:*

$$R_{hinge}(\mathbf{t}, \mathbf{v}, \tilde{r}) \to R_{hinge}^* \implies R_{01c}(\varphi(\mathbf{t}, \mathbf{v}, \tilde{r})) \to R_{01c}^*, \tag{15}$$

*i.e., the surrogate loss $\ell_{hinge}^c(\mathbf{v}, \mathbf{t}, r, x, z, y)$ 11 is consistent with respect to the loss of interest $\ell_{01c}(\varphi(\mathbf{t}, \mathbf{v}, r), x, z, y)$.*

The proof is provided in Appendix A.5. The proof is built by showing that 1) the minimizers of both risks are unique and coincide:

$$f_{hinge}^* = f^*, \tag{16}$$

(Lemma A.1, with proof provided in Appendix A.5.1); and 2) that for some increasing function $\Psi$ with $\Psi(0) = 0$, the following holds:

$$R_{01c}(\varphi(\mathbf{t}, \mathbf{v}, \tilde{r})) - R_{01c}^* \leq \Psi\left( R_{hinge}(\mathbf{t}, \mathbf{v}, \tilde{r}) - R_{hinge}^* \right), \tag{17}$$

(Lemma A.2, with proof included in Appendix A.5.2). Taken together, these results guarantee consistency. We can actually establish that

$$R_{01c}(\varphi(\mathbf{t}, \mathbf{v}, \tilde{r})) - R_{01c}^* \leq \frac{2(K-1)}{K} \left( R_{hinge}(\mathbf{t}, \mathbf{v}, \tilde{r}) - R_{hinge}^* \right). \tag{18}$$

The bound on the risk gap provided by (18) allows us to further quantify the relationship between the two optimization problems, showing that the consistency is not merely asymptotic. This upper bound is tight and attainable for some cases of $\eta$ and $\zeta$. The $\frac{K-1}{K}$ term comes from the scaling of the multi-hinge loss, while the factor of 2 accounts for corner cases where the routing decision is uncertain ($\tilde{r}(x) = 0.5$) and the model perfectly estimates the posteriors $\eta$ and $\zeta$.

## 4.1 Cross entropy version

One might be tempted to build a similar formulation using the widely used cross-entropy loss $-\log(\mathbf{p}_y)$. Some heuristics in the literature for training two-stage or early exit models are built around a similar version of this loss [Regol et al., 2024]. Interestingly, we can prove that such a loss is in fact not Bayes consistent with the $0 - 1c$ loss that we presented. To build a cross entropy version of the proposed loss, we now need to assume that the model outputs predicted class probabilities $\mathbf{p}^1 \in \Delta^K$ for $f_1$ and $\mathbf{p}^2 \in \Delta^K$ for $f_2$, where $\Delta^K$ is the $K-$dimensional simplex and $\varphi$ is the same link function that was previously defined. The cross entropy version of the loss that we consider adds an arbitrary function of the cost $g(c)$ and is given by:

$$\ell_{ce}^{g(c)}(\mathbf{p}^1, \mathbf{p}^2, \tilde{r}, x, z, y) = -\left((1 - \tilde{r}(x)) \log(\mathbf{p}_y^1) + \tilde{r}(x)(\log(\mathbf{p}_y^2) + g(c))\right). \quad (19)$$

We again consider the associated risk:

$$R_{ce}(\mathbf{p}^1, \mathbf{p}^2, \tilde{r}) = \mathbb{E}_{p(x,z,y)}[\ell_{ce}^{g(c)}(\mathbf{p}^1, \mathbf{p}^2, \tilde{r}, x, z, y)], \quad (20)$$

and the minimizing function:

$$f_{ce}^* = \underset{\mathbf{p}^1, \mathbf{p}^2, \tilde{r} \in [0,1]}{\arg\min} R_{ce}(\mathbf{p}^1, \mathbf{p}^2, \tilde{r}). \quad (21)$$

The following lemma shows that this coss-entropy surrogate loss cannot be Bayes-consistent.

**Lemma 4.2.** *There is no function $g(\cdot)$ for which the solution $f_{ce}^*$ to the associated problem in Eqn. 21 is equal to the Bayes-classifier $f^*$ defined in Eqn. 6 for all distributions $p(X, Z, Y)$.*

The proof is included in Appendix A.6.

# 5 Experiments

## 5.1 Synthetic Experiments

To validate our findings, we present a synthetic experiment in which the ground-truth posteriors are known. We design a simple $K$-class classification task with one-dimensional inputs $X$ and $Z$ to enable visualization of the learned functions. Our primary interest lies in visualizing the decision boundary $\tilde{r}(x)$ of a model $f$ trained with the proposed surrogate loss. This boundary should closely approximate the optimal decision rule $r^*(x)$, as defined in Eqn. 9. For completeness, we additionally include an experiment using the related learning-to-defer baselines, which we adapt to this particular setting in Appendix A.3.

**Task Description** The inputs $X$ and $Z$ are drawn uniformly from the interval $[-1, 1)$. The label $Y$ is sampled from a categorical distribution with parameter $\boldsymbol{\theta} = [\theta_1, \theta_2, \ldots, \theta_K]^T \in [0, 1]^K$, where $\sum_{i=1}^K \theta_i = 1$ and $p(Y = i) = \theta_i$. The function $\boldsymbol{\theta}(x, z)$ is defined piecewise by partitioning the domain of $x, z$ into $K - 1$ slanted regions. Full details of the construction of the synthetic dataset are provided in Appendix A.1.1. The random variables are distributed as:

$$X \sim \text{Uniform}[-1, 1) = p(X) \quad (22)$$

$$Z \sim \text{Uniform}[-1, 1) = p(Z) \quad (23)$$

$$Y \sim \text{Categorical}(\boldsymbol{\theta}(x, z)) = p(Y|x, z) \quad (24)$$

The constructed task can be visualized in Figure 1, where we show the class distribution in terms of most likely class and the samples $x_i, y_i, z_i \sim p(X, Z, Y)$ for $K = 5$. For this example, we can see that at $x = 0$, the value of $z$ provides essentially no additional information to estimate the correct posterior, which should translate into no deferral to $f_2$ ($r^*(x = 0) = 0$). At $x = 0.25$, the variable $z$ becomes informative. Therefore, the optimal decision function $r^*(x)$ will alternate as vertical strips along the x-axis, with width of size that varies based on the cost parameter $c$.

Given this construction, the exact posterior probabilities can be computed in closed form, allowing us to derive the optimal decision rule $r^*(x)$. To approximate the expectations $\mathbb{E}_{p(Z|x)}$, we use Monte Carlo estimation by sampling from $p(Z|x)$:

$$\hat{r}^*(x) \approx \begin{cases} 0 & \text{if } \max_{y \in \mathcal{Y}}[\frac{1}{M} \sum_{i=1}^M \zeta_y(x, z_i)] \geq \frac{1}{M} \sum_{j=1}^M \max_{y \in \mathcal{Y}} \zeta_y(x, z_j) - c \\ 1 & \text{otherwise} \end{cases} \quad (25)$$

$$\text{where } z_i, z_j \sim \text{Uniform}[-1, 1) \quad (26)$$

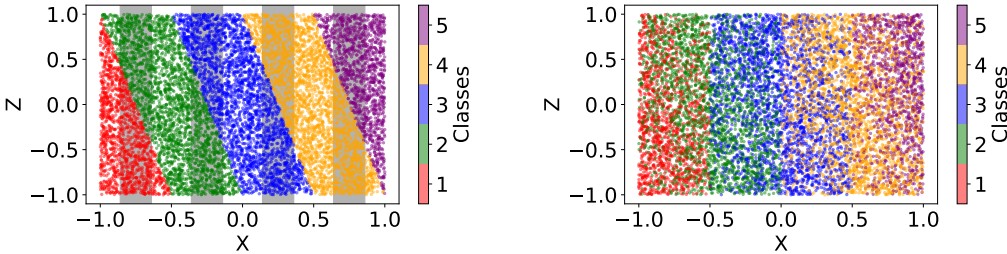

Figure 1: Visualization of multi class synthetic dataset with $k = 5$. **Left)** Max probability labels $\arg\max_y \quad p(Y = y | X, Z)$. The black shaded region indicates where the optimal decision rule is to defer to $f_2$ ($r^*(x) = 1$), given a cost of $c = 0.03$. **Right)** Samples $x, z, y \sim p(X, Z, Y)$ of the synthetic experiment.

In our experiments, we use $M = 1000$ samples to approximate the expectations.

**Training details** We build simple 3-layer neural networks (NN) for $\mathbf{t}$, $\mathbf{v}$, and $\tilde{r}$. Following the requirements for $\tilde{r}(x)$, the corresponding network takes $x$ as input and ends with a sigmoid activation. For $\mathbf{t}$ and $\mathbf{v}$, the NNs take $x$ and $(x, z)$ as inputs, respectively, and output a real-valued vector of dimension $K-1$. Appendix A.1.2 provides parameter size and layer details. We use the Adam optimizer with learning rate $lr = 0.001$, a batch size of 512 and train for 50 epochs using our surrogate loss defined in Eqn. 11. The training set size is $N_{tr} = 10,000$ and the test set size is $N_{te} = 1,000$.

**Result and discussion** Figure 2 illustrates the ground truth and predicted decision boundaries for cost values $c = 0.03, 0.07$ and number of classes $K = 3, 5$. We observe that the model trained with the proposed surrogate loss successfully learns the correct decision boundary across different cost values and numbers of classes $K$. The learned decision function $\tilde{r}(x)$ perfectly tracks with the ground truth $r^*(x)$. Additionally, although the trained model can output any value in the range $\tilde{r}(x) \in [0, 1]$ due to the sigmoid activation, it learns to produce sharp values near 0 or 1, which is the optimal behavior.

Now if we turn to a model trained with the additive version of the cross-entropy-based surrogate loss introduced in Eqn 19, using the identity function $g(c) = c$ with $K = 5$, we observe in Figure 3 that the behavior of the learned decision function $\tilde{r}(x)$ differs significantly. First, we note that since consistency cannot be established for this surrogate loss, it is not possible to precisely target a desired cost level in the $\ell_{01c}$ loss, unlike with the hinge-based surrogate. Looking at the results, the learned decision boundaries are generally unstable and uneven. The correct pattern of deferral for $K = 5$ can be observed in the top-right plot of Figure 2, where we see that four regions should be evenly spaced out and deferred to $f_2$ (regardless of $c$). This pattern is not adequately learned in Figure 3. For instance, we see that the right-most region of $x$ that should be deferred is slowly erased as the cost increases.

Lastly, we visualize the behavior of the learned model $f_{\text{hinge}}$ during training in Figure 4. We track the empirical target risk estimated from sampling $\hat{R}_{01c}(f) = \frac{1}{N} \sum_{i=1}^{N} \ell_{01c}(f(x_i, z_i), y_i)$ and observe that it converges to the (empirical) optimal risk $\hat{R}_{01c}^*$ as expected.

## 5.2 Large Language Model Experiment

To illustrate a practical setting of the problem we consider, we present an experiment based on large language models (LLMs). In this experiment, we use two LLMs of different sizes, which correspond to different inference costs. The additional inference cost used by the larger model corresponds to $c$ in our setup. The task involves solving multi-answer math questions. The intuition behind this setting is that some test questions should be more difficult than others. Therefore, it would be desirable to efficiently dispatch simpler questions to the smaller LLM and more challenging ones to the larger LLM. This allows us to achieve strong performance at a reasonable inference cost.

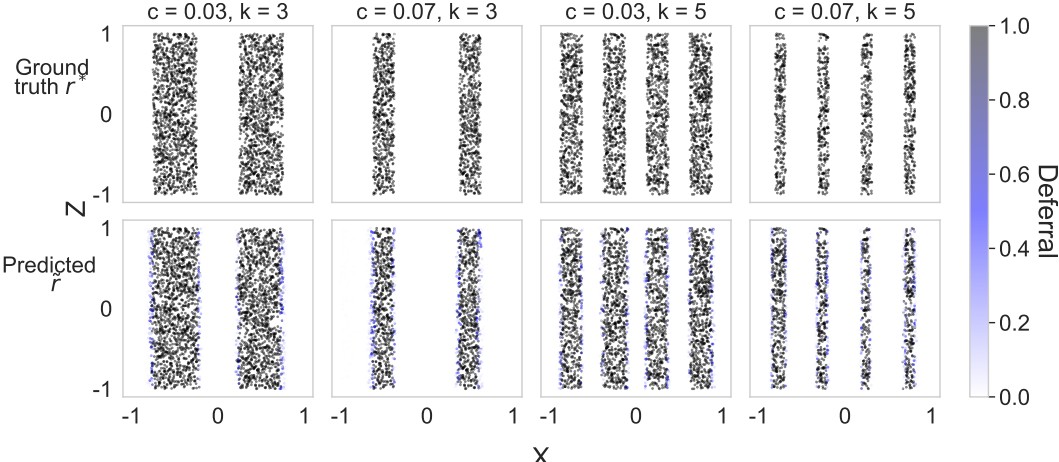

Figure 2: **Top)** Ground truth decision boundary $\hat{r}^*(x)$ with 2 costs values $c = 0.03, 0.07$ and number of classes $K = 3, 5$. **Bottom)** Learned $\tilde{r}(x)$ of the model that was trained with our surrogate loss. In all cases, the two decision boundaries are perfectly aligned, confirming our result that the model trained with the proposed surrogate loss successfully learns the optimal decision function. As the cost increases, the black region which represents points deferred to $f_2$ shrinks.

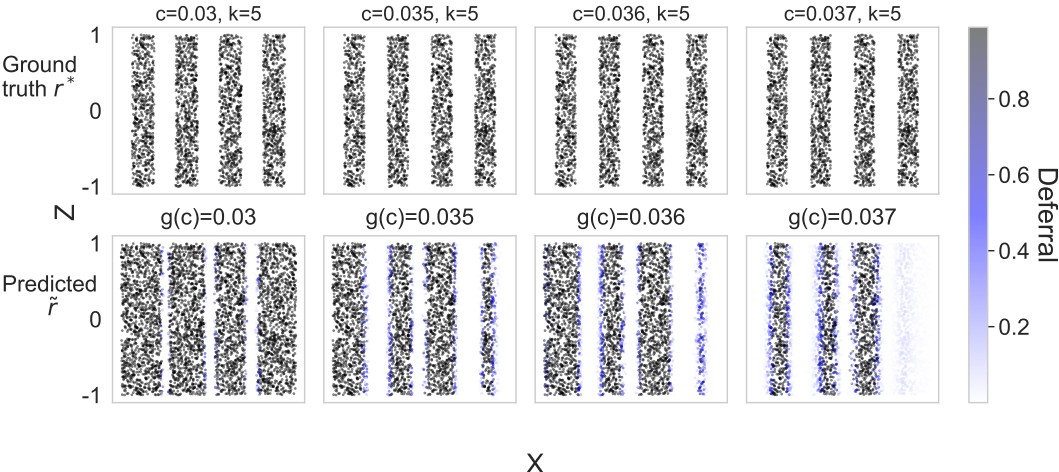

Figure 3: **Top)** Ground truth decision boundary $\hat{r}^*(x)$ with 4 costs values and number of classes $K = 5$. **Bottom)** Learned $\tilde{r}(x)$ of a model trained with the additive cross-entropy surrogate loss with $g(c) = c$, for varying $c$ values. Unlike the model trained with the hinge-based surrogate, the learned decision patterns are generally wrong and not consistent.

**Task description**  We use the Instruction-Tuned Pre-trained models LLaMA 3 8B and LLaMA 3 70B [Grattafiori et al., 2024] to solve multi-answer math questions from the AQUA dataset [Zhong et al., 2024]. The AQUA dataset is composed of multiple-choice math reasoning questions, each with 5 choices. We frame the task as a 5-class classification problem, where the model must select the correct option from a fixed set. The inputs $x$ and $z$ are formed by extracting the hidden-states from the final tokens of the 8B LLM and the 70B LLM, respectively. We use the first 1000 AQUA [Zhong et al., 2024] datapoints from the test split as our dataset, and use a $80/10/10$ train/val/test split.

**Training details**  We use a similar architecture to the one previously presented. The model is trained for 1000 epochs using a learning rate of 0.001, a batch size of 32, and early stopping with a patience of 20 epochs. Additional details are provided in Appendix A.1.3.

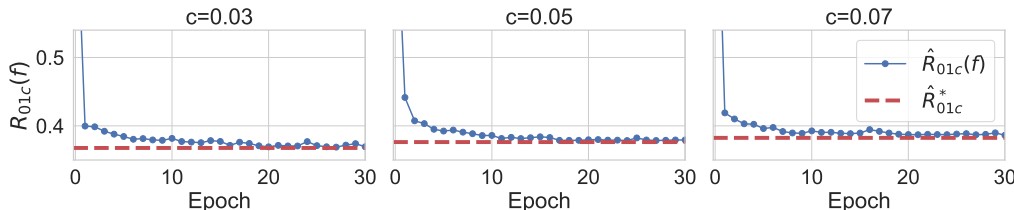

Figure 4: Empirical $0 - 1c$ risks of the learned function trained with the surrogate loss and of the optimal solution for $K = 5$. We can see that for varying cost values, function trained with the surrogate loss converges to the optimal solution.

**Results and discussion** Although we do not have access to the ground truth decision function in this setting, we can examine the accuracy evaluated on the selected samples vs. all samples of the model trained with the surrogate loss. The selected samples of $f_1$ or $f_2$ are the samples routed to these functions by $\tilde{r}(x)$. Ideally, the two-stage classifier model $f$ should learn to route "hard" examples to $f_2$, and "easier" examples to $f_1$. In practice, the surrogate loss can have two effects: 1) $f_1$ and $f_2$ are additionally trained on their respective selected samples; and 2) $f_1$ and $f_2$ may receive smaller gradient updates depending on the average deferral rates.

These two effects can be observed in Figure 5. In the left figure tracking $f_1$, we see that the average accuracy slightly increases as the cost increases (and consequently the deferral rate decreases), and the inverse behavior can be seen for $f_2$ in the right figure. $f_1$ should, in principle, be given an easier task, so we can expect its selected accuracy to be higher than the average accuracy, which we observe in the left panel of Figure 5. The two values are closest when the selected samples comprise almost all the data (i.e., a deferral rate of $90\%$). For $f_2$, the selected accuracy is closer to the average accuracy. This could suggest that training only on the samples deferred to $f_2$ does not result in better performance—possibly because these consist of "harder" instances.

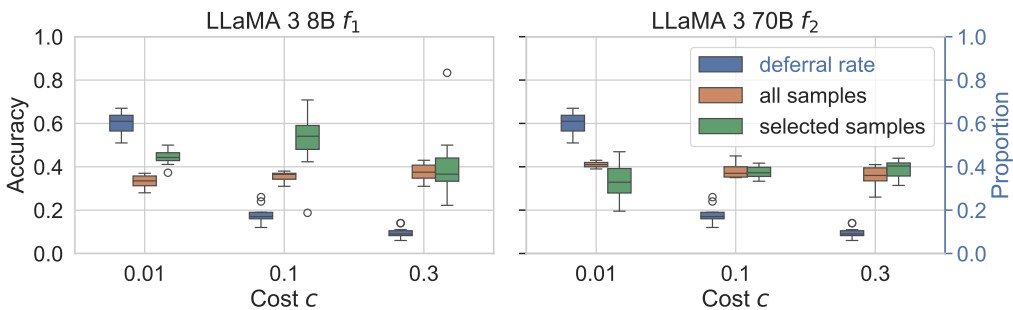

Figure 5: Deferral rate and average accuracy on all samples and on selected samples by **left)** LLaMA 3 8B $f_1$ and by **right)** LLaMA 3 70B $f_2$. The confidence intervals are computed on 10 trials.

In addition to aggregate performance, we can also inspect which types of queries are routed to each model. Figure 6 shows examples of math questions that were consistently routed to the smaller model ($f_1$) and the larger model ($f_2$) across various cost settings. From the presented examples, it appears that the "easy" questions that were consistently routed to the small LLM ($f_1$) generally involve basic arithmetic or proportions. In contrast, the labeled "hard" questions that were consistently routed to the large LLM ($f_2$) seem to require more comprehensive knowledge (such as motion or number theory). This suggests that the routing function aligns with our perceived notion of complexity and the type of reasoning required. See Appendix A.2 for the complete list of questions that were consistently routed to $f_1$ and $f_2$.

| **Example questions routed to the small LLM ($f_1$)** |
|---|
| **Question 1:** The cost of 10 kg of mangos is equal to the cost of 24 kg of rice. The cost of 6 kg of flour equals the cost of 2 kg of rice. The cost of each kg of flour is $22. Find the total cost of 4 kg of mangos, 3 kg of rice and 5 kg of flour? 
 **Question 2:** A man buys an article and sells it at a profit of 20%. If he had bought it at 20% less and sold it for Rs.75 less, he could have gained 25%. What is the cost price? |

| **Example questions routed to the larger LLM ($f_2$)** |
|---|
| **Question 1:** Two trains 140 m and 160 m long run at the speed of 60 km/hr and 40 km/hr respectively in opposite directions on parallel tracks. The time which they take to cross each other is? 
 **Question 2:** If the product of two numbers is 17820 and their H.C.F. is 12, find their L.C.M. |

Figure 6: Sampled questions that are consistently being routed to $f_1$ or $f_2$ across different costs.

## 6 Conclusion and Limitations

In conclusion, this work aims to solidify the theoretical foundation behind the design and use of loss functions for the increasingly relevant problem of training multiple models with different costs, while also learning which model to use. We formalized this problem using a principled $0 - 1$ cost-based loss formulation and proposed a surrogate loss based on the hinge loss, showing its consistency.

**Limitations**  A clear limitation of our work is that we only consider two models in our setup, whereas dynamic networks often require more than two. Extending our approach to the multi-stage setting would be a valuable direction for future research. Appendix A.7 provides a sketch of how our method can be generalized to the multi-stage setting with $L$ classifiers. Moreover, although the theoretical results guarantee loss consistency, the hinge loss is less commonly used in practice. While we have presented a simple proposal of a cross-entropy surrogate loss and shown that it is insufficient for this setting, exploring alternative, more stable losses would be an important next step to ensure the development of practical and principled methods.

## 7 Social Impact

Although we believe that this theoretical paper poses minimal direct societal impact, the broader problem of cost-sensitive deferral systems may raise concerns related to fairness and access. In such systems, the model determines whether a query is "simple" and can be handled by a smaller model, or "difficult" and should be deferred to a more powerful model, which may involve higher computational cost or latency. This can introduce bias in how different users' queries are treated. For instance, if a particular user or group systematically submits queries that the system deems "hard", they may consistently experience greater latency, potentially leading to unfair treatment or limited access. Additionally, this introduces new potential pathways for bias to enter the system, as the deferral rule itself can be biased. This could further exacerbate disparities in user experience and overall system fairness.

## Acknowledgements

We thank Prof. Rui Pires da Silva Castro for his valuable insights and suggestions, which greatly contributed to the development of our solution. This research was partially funded by the Natural Sciences and Engineering Research Council of Canada (NSERC), [reference number 260250]. Cette recherche a été partiellement financée par le Conseil de recherches en sciences naturelles et en génie du Canada (CRSNG), [numéro de référence 260250]. Ce projet de recherche nᵒ 324302 est rendu possible grâce au financement du Fonds de recherche du Québec.

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

# A Appendix


### A.1 Additional experimental details

#### A.1.1 Synthetic task

In this section, we provide additional details of the synthetic task. We model $Y$ by a categorical distribution with parameter $\boldsymbol{\theta} = [\theta_1, \theta_2, \ldots, \theta_k]^T \in [0,1]^k$ satisfying $\sum_{i=1}^k \theta_i = 1$ and $p(Y = i) = \theta_i$. $\boldsymbol{\theta}$ is modeled as a piecewise function by partitioning the range of $X$ into $k-1$ equally sized bins $\{B_1, B_2, \ldots, B_{k-1}\}$. Assuming the range of $X$ is $[a, b), a, b \in \mathbb{R}$, we define $B_i = \left[ a + \frac{(i-1)(b-a)}{k-1}, a + \frac{(i)(b-a)}{k-1} \right)$. Within every bin $B_i$, only $\theta_i$ and $\theta_{i+1}$ take on non-zero values, following a scaled and shifted sigmoid:

$$X \sim Uni[-1, 1] \tag{27}$$

$$Z \sim Uni[-1, 1] \tag{28}$$

$$Y \sim Categorical(\boldsymbol{\theta}) \tag{29}$$

$$\boldsymbol{\theta} = [\theta_1, \theta_2, \ldots, \theta_k]^T \in [0, 1]^k \tag{30}$$

$$B_i = \left[ -1 + \frac{2(i-1)}{k-1}, -1 + \frac{2i}{k-1} \right) \forall i \in [1, 2, \ldots, k-1] \tag{31}$$

$$\theta_i = \begin{cases} \sigma(sX + c_i + Z) & \text{if } X \in B_i \\ \sigma(-1 \times (sX + c_{i-1} + Z)) & \text{if } i > 1 \text{ and } X \in B_{i-1} \\ 0 & \text{else} \end{cases} \tag{32}$$

$$s = k - 1 \tag{33}$$

$$c_i = k - 2i \tag{34}$$

$$\tag{35}$$

Using this model, we arrive at the closed form for the posteriors:

$$\eta_{y=i}(x) = \int p(Y = i | x, Z) p(Z|x) dZ = \mathbb{E}_{p(Z)}[\theta_i] \tag{36}$$

$$= \mathbb{E}_{p(Z)} \left[ \begin{cases} \sigma(sx + c_i + Z) & \text{if } x \in B_i \\ \sigma(-1 \times (sx + c_{i-1} + Z)) & \text{if } i > 1 \text{ and } x \in B_{i-1} \\ 0 & \text{else} \end{cases} \right], \tag{37}$$

$$\zeta_{y=i}(x, z) = \theta_i = \left[ \begin{cases} \sigma(sx + c_i + z) & \text{if } x \in B_i \\ \sigma(-1 \times (sx + c_{i-1} + z)) & \text{if } i > 1 \text{ and } x \in B_{i-1} \\ 0 & \text{else} \end{cases} \right], \tag{38}$$

and we can derive the optimal decision function $r^*(x)$:

$$r^*(x) = \begin{cases} 0 & \text{if } \max_i \mathbb{E}_{p(Z)}[\theta_i] \geq \mathbb{E}_{p(Z)}[\max_i \theta_i] - c, \\ 1 & \text{o.w.} \end{cases} \tag{39}$$

#### A.1.2 Synthetic Model details

We describe the architecture of the model used in the synthetic experiment. The hidden size of all networks is 64. Each neural network is defined as:

$$r_\theta(x) = sigmoid\left(BatchNorm\left(\Theta ReLU(BatchNorm(\boldsymbol{\theta}x))\right)\right) \tag{40}$$

$$\mathbf{t}_\theta(x) = \Theta ReLU(BatchNorm(\Theta ReLU(BatchNorm(\boldsymbol{\theta}x)))) \tag{41}$$

$$\mathbf{v}_\theta(x, z) = \Theta ReLU(BatchNorm(\Theta ReLU(BatchNorm(\Theta[x, z])))). \tag{42}$$

#### A.1.3 LLM Model details

We describe the architecture of the model used in the LLM experiment. The hidden size of all networks is 128. We performed a grid search for the hidden size across the values $\{32, 64, \underline{128}, 256\}$ and for the learning rate across the values $\{0.01, \underline{0.001}, 0.0001\}$.

Each neural network is defined as:

$$r_\theta(x) = sigmoid\left(BatchNorm\left(\Theta ReLU(BatchNorm(\Theta x))\right)\right) \tag{43}$$

$$\mathbf{t}_\theta(x) = \Theta ReLU(BatchNorm(\Theta ReLU(BatchNorm(\Theta x)))) \tag{44}$$

$$\mathbf{v}_\theta(x, z) = \Theta ReLU(BatchNorm(\Theta ReLU(BatchNorm(\Theta[x, z])))). \tag{45}$$

## A.2 Complete list of deferred questions (LLM experiments)

In this section, we provide a comprehensive list of the questions that were consistently deferred to $f_2$ or sent to $f_1$ in the LLM experiment.

**Example questions sent to the small LLM ($f_1$):**

1. A man buys an article and sells it at a profit of 20%. If he had bought it at 20% less and sold it for Rs. 75 less, he could have gained 25%. What is the cost price?

2. The cost of 10 kg of mangos is equal to the cost of 24 kg of rice. The cost of 6 kg of flour equals the cost of 2 kg of rice. The cost of each kg of flour is $22. Find the total cost of 4 kg of mangos, 3 kg of rice and 5 kg of flour?

3. The speed of a boat in upstream is 100 kmph and the speed of the boat downstream is 180 kmph. Find the speed of the boat in still water and the speed of the stream.

4. A and B working together could mow a field in 28 days and with the help of C they could have mowed it in 21 days. How long would C take by himself?

5. Evaluate the expression:
$$2^2 + 4^2 + 6^2 + \ldots + 22^2$$

6. In an examination, 60% failed in Math and 40% failed in French. If 15% failed in both subjects, what percentage of students passed in both?

7. One train crosses a bridge of length 340 m in 42 seconds, and the same train crosses another bridge of length 500 m in 50 seconds. What is the approximate speed of the train in km/hr?

8. Eshan and Mary each wrote two or three poems every day over a period of time. Eshan wrote 43 poems while Mary wrote 61. What is the number of days in this period?

9. Find the value of $x$ in the sequence of numbers 5, 1, 6, 0, 4, 8, $x$, 2 if the sum of the first 7 numbers is 30 and the average is 4.

10. Roja and Pooja start moving in opposite directions from a pole. They are moving at speeds of 7 km/hr and 3 km/hr respectively. After 4 hours, what will be the distance between them?

**Example questions deferred to the larger LLM ($f_2$):**

1. If the product of two numbers is 17820 and their H.C.F. is 12, find their L.C.M.

2. Two passenger trains start at the same hour in the day from two different stations and move towards each other at the rate of 14 kmph and 21 kmph respectively. When they meet, it is found that one train has traveled 60 km more than the other one. What is the distance between the two stations?

3. Which is the odd one: 10, 25, 45, 54, 60, 75, 80?

4. Two trains 140 m and 160 m long run at the speed of 60 km/hr and 40 km/hr respectively in opposite directions on parallel tracks. The time which they take to cross each other is?

5. A ladder 100 feet long is leaning against a vertical wall. Its lower end is 60 feet from the bottom of the wall. The side of the largest cubical box that can be placed between the wall and the ladder without disturbing the ladder is (to the nearest foot)?

6. On dividing a certain number by 5, 7 and 8 successively, the remainders obtained are 2, 3 and 4 respectively. When the order of division is reversed and the number is successively divided by 8, 7 and 5, what will be the respective remainders?

7. A tour group of 25 people paid a total of $670 for entrance to a museum. If this price included a 5% sales tax, and all the tickets cost the same amount, what was the face value of each ticket price without the sales tax?

8. A rectangular floor is covered by a rug except for a strip 4 meters wide along each of the four edges. If the floor is 25 meters by 20 meters, what is the area of the rug in square meters?

9. In each of the following questions a number series is given with one term missing. Choose the correct alternative that will continue the same pattern and fill in the blank space.

$$2, \ 7, \ 14, \ ?, \ 34, \ 47$$

10. In a game of 500 points there are three participants A, B, and C. A gives to B 80 points and to C 101 points. Then how many points can B give to C?

11. When magnified 1,000 times by an electron microscope, the image of a certain circular piece of tissue has a diameter of 2 centimeters. The actual diameter of the tissue, in centimeters, is:

12. From the given equation, find the value of $x$:

$$2x^2 + 9x - 5 = 0$$

13. The sum of the non-prime numbers between 50 and 60, non-inclusive, is:

14. Solve the system of equations to find the values of $c$ and $d$:

$$\text{I. } c^3 - 988 = 343 \tag{46}$$

$$\text{II. } d^2 - 72 = 49 \tag{47}$$

15. How many minutes does Aditya take to cover a distance of 400 meters, if he runs at a speed of 20 km/hr?

16. An engineer designed a ball so that when it was dropped, it rose with each bounce exactly one-half as high as it had fallen. The engineer dropped the ball from an 18-meter platform and caught it after it had traveled 53.4 meters. How many times did the ball bounce?

## A.3 Additional baselines

We include additional baselines that assume different settings from ours, such as the learning-to-defer-to-expert setting, which trains a classifier alongside a deferral function that defers to a fixed expert, as in [Mozannar and Sontag, 2020]. We also consider simple thresholding methods that only produce results for deferral, assuming fixed classifiers, including the popular confidence-based thresholding rule investigated by Jitkrittum et al. [2023]. These methods generally rely on training cross-entropy models for $f_1$, and obtain the associated predicted probabilities $\mathbf{p}_1$. To adapt these baselines to our setting, we train the second classifier $f_2$ separately using a standard cross-entropy loss, and then follow the scheme of the baselines to train the first classifier $f_1$ and obtain the decision function $\tilde{r}(x)$. Let $\hat{acc}_f$ denote the empirical accuracy of a model $f$ evaluated on a validation set.

We include the following rules:

**CT-c:** We pretrain $f_1$ using cross-entropy. For a given cost $c$, we set the threshold $\tau$ using the empirical accuracy of the second classifier minus the cost $\tau = \hat{acc}_{f_2} - c$: and define the deferral rule as:

$$\tilde{r}(x) = \begin{cases} 0 & \text{if } \max_{y \in \mathcal{Y}} \mathbf{p}_y^1 \geq \tau, \\ 1 & \text{otherwise} \end{cases} \tag{48}$$

**Soft deferral:** We pretrain $f_1$ using cross-entropy, and sample the deferral decision from a Bernouilli distribution with $1 - \max_{y \in \mathcal{Y}} \mathbf{p}_y^1$ as a parameter:

$$\tilde{r}(x) \sim \text{Bernoulli}\big(1 - \max_{y \in \mathcal{Y}} \mathbf{p}_y^1\big). \tag{49}$$

**CT:** We pretrain $f_1$ using cross-entropy, and search for the optimal threshold $\tau$ that yields the smallest empirical risk on a validation set.

$$\tilde{r}(x) = \begin{cases} 0 & \text{if } \max_{y \in \mathcal{Y}} \mathbf{p}_y^1 \geq \tau, \\ 1 & \text{otherwise} \end{cases} \tag{50}$$

**L2D:** We use our the pretrained $f_2$ as the expert for the method of Mozannar and Sontag [2020]. We set the confidence parameter in our expert, $\alpha$, to be the average accuracy of $f_2$: $\alpha = a\hat{c}c_f$.

Figure 7 shows the empirical risk during training for a setting with a cost of $c = 0.3$ and number of classes $k = 5$. We observe that our proposed approach converges the fastest and achieves the lowest empirical risk. In Table 1, we report the empirical risk $\hat{R}_{01c}$ along with the standard deviation computed across 10 trials for varying costs. Our proposed surrogate loss consistently attaining the lowest empirical risk across all cost settings.

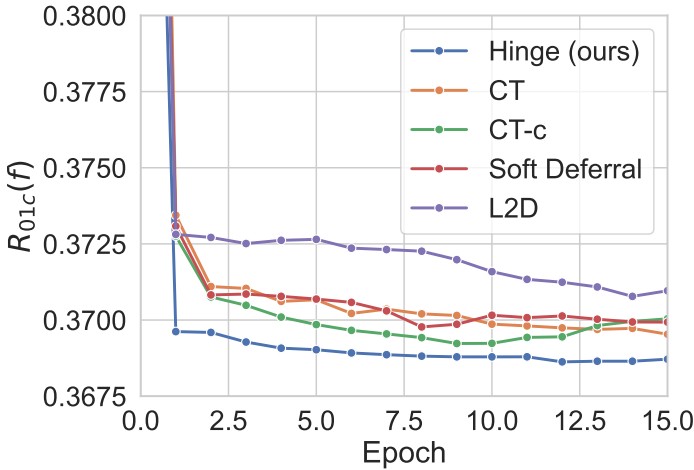

Figure 7: Empirical risk $\hat{R}_{01c}$ of the learned function trained with our proposed surrogate loss and other baselines for $K = 5$ and $c = 0.3$.

Table 1: Empirical risk $\hat{R}_{01c}$ for different baselines and deferral costs $c$ with $K = 5$. The mean and standard deviation are computed across 10 trials. The bolded entry denotes the lowest value.

| Baseline | $c = 0.03$ | $c = 0.05$ | $c = 0.07$ |
|---|---|---|---|
| CT-c | $0.3701 \pm 0.0013$ | $0.3842 \pm 0.0007$ | $0.4031 \pm 0.0014$ |
| CT | $0.3700 \pm 0.0008$ | $0.3784 \pm 0.0016$ | $0.3853 \pm 0.0036$ |
| Soft deferral | $0.3700 \pm 0.0012$ | $0.3789 \pm 0.0008$ | $0.3855 \pm 0.0014$ |
| L2D | $0.3712 \pm 0.0015$ | $0.3826 \pm 0.0022$ | $0.3906 \pm 0.0025$ |
| Hinge (ours) | $\mathbf{0.3695 \pm 0.0004}$ | $\mathbf{0.3777 \pm 0.0004}$ | $\mathbf{0.3842 \pm 0.0005}$ |

## A.4 Proof of the solution $f^*$

In this section, we provide the proof of Lemma 3.1, which states that the optimal solution

$$f^* = \arg\min_{f \in \mathcal{H}} R_{01c}(f) \tag{51}$$

is the following:

$$f^* = \begin{cases} \arg\max_y \eta_y(x), & \text{if } \max_y \eta_y(x) \geq \mathbb{E}_{p(Z|x)}[\max_y \zeta_y(x, Z)] - c, \\ \arg\max_y \zeta_y(x, z), & \text{else}. \end{cases} \tag{52}$$

Or alternatively:

$$f^* = \begin{cases} f_1^*(x), & r^*(x) = 0 \\ f_2^*(x,z), & r(x) = 1 \end{cases},$$ (53)

$$\text{where } f_1^*(x) = \arg\max_{y \in \mathcal{Y}} \eta_y(x), \quad \forall x \ s.t. \ r^*(x) = 0$$ (54)

$$f_2^*(x,z) = \arg\max_{y \in \mathcal{Y}} \zeta_y(x,z), \quad \forall x \ s.t. \ r^*(x) = 1$$ (55)

$$r^*(x) = \mathbb{1}\left[ max_y \eta_y(x) \leq \mathbb{E}_{p(Z|x)}[\max_y \zeta_y(x,Z)] - c \right].$$ (56)

*Proof.* We start by evaluating the risk w.r.t to the function:

$$f^s = \begin{cases} \arg\max_y \eta_y(x) & \text{if } \max \eta(x) \geq \mathbb{E}_{p(z|x)} \max \zeta(x,z) - c \\ \arg\max_y \zeta_y(x,z) & \text{else,} \end{cases}$$ (57)

$$R_{01c}(f^s) = \mathbb{E}_{p(x,z,y)}[\ell_{01c}(f^s(x,z), y)],$$ (58)

and prove the result by showing that any function $f^o \neq f^s$ results in a higher risk, therefore showing that $f^s$ is the optimal solution.

The risk is given by:

$$R_{01c}(f^s) = \mathbb{E}_{p(x,z,y)}[\ell_{01c}(f^s(x,z), y)]$$ (59)

$$R_{01c}(f^s) = \int_x \int_z \sum_y [\ell_{01c}(f^s(x,z), y)] p(y|x,z) p(z|x) p(x) dz dx.$$ (60)

We can partition $\mathcal{X}$ in two regions based on the decision function of $f^s$, i.e. : $A = \{x; \max \eta(x) \geq \mathbb{E}_{p(z|x)} \max \zeta(x,z) - c\}$ and $B = \{x; \max \eta(x) \leq \mathbb{E}_{p(z|x)} \max \zeta(x,z) - c\}$ and split the expectation in two terms:

$$R_{01c}(f^s) = R^A + R^B$$ (61)

$$\text{where } R^S \triangleq \int_{x \in S} \int_z \sum_y [\ell_{01c}(f^s(x,z), y)] p(y|x,z) p(z|x) p(x) dx dz.$$ (62)

Looking at both terms separately, starting with $R^A$ where $f^s$ does not use $z$ (or corresponds to $f_1$):

$$R^A = \int_{x \in A} \int_z \sum_y [\ell_{01c}(f^s(x,z), y)] p(y|x,z) p(z|x) p(x) dx dz$$ (63)

$$= \int_{x \in A} \sum_y \left( \int_z \mathbb{1}[f_1(x) \neq y] p(z|x,y) dz \right) p(x,y) dx \text{ by def of } f^s \text{ and } \ell_{01c}$$ (64)

$$= \int_{x \in A} \sum_y \mathbb{E}_{z|x,y} \left[ \mathbb{1}[f_1(x) \neq y] \right] p(x,y) dx$$ (65)

$$= \int_{x \in A} \sum_y \mathbb{1}[f_1(x) \neq y] p(x,y) dx \text{ as nothing depends on } z$$ (66)

$$= \int_{x \in A} \sum_y \mathbb{1}[\arg\max_{y'} \eta_{y'}(x) \neq y] \eta_y(x) p(x) dx \text{ by def of } f_1^s \text{ and } p(y|x)$$ (67)

$$R^A = \int_{x \in A} 1 - \max_y \eta_y(x) p(x) dx.$$ (68)

We can obtain more straightforwardly $R^B$:

$$R^B = \int_{x \in B} \int_z (1 + c - \max_y \zeta_y(x,z) p(x,z)) dz dx.$$ (69)

Hence, the risk of $f^s$ is given by;

$$R_{01c}(f^s) = \int_{x \in A} 1 - \max_y \eta_y(x) p(x) dx + \int_{x \in B} \int_z 1 + c - \max_y \zeta_y(x,z) p(x,z) dz dx.$$ (70)

Now, we consider a different two-stage classifier $f^o \in \mathcal{H}$ and $f^s \neq f^o$. We show that any $f^o \in \mathcal{H}$ will lead to a higher risk $R_{01c}(f^s) \leq R_{01c}(f^o)$, therefore proving that $f^s = f^*$.

We further partition the space $\mathcal{X}$ by splitting $A$ and $B$ where $f^s \neq f^o$ and $f^s = f^o$

$$A^s \triangleq \{x; x \in A \text{ and } f^s = f^o\}, \qquad A^d \triangleq \{x; x \in A \text{ and } f^s \neq f^o\}, \tag{71}$$

$$B^s \triangleq \{x; x \in B \text{ and } f^s = f^o\}, \qquad B^d \triangleq \{x; x \in B \text{ and } f^s \neq f^o\}. \tag{72}$$

We then use those new partitions to further decompose the risks of $f^s$ and $f^o$ in 4 terms:

$$R_{01c}(f^s) = R^{A^s} + R^{A^d} + R^{B^s} + R^{B^d} \tag{73}$$

$$\text{and } R_{01c}(f^o) = R_o^{A^s} + R_o^{A^d} + R_o^{B^s} + R_o^{B^d}. \tag{74}$$

We can then write the difference between the risks as:

$$R_{01c}(f^s) - R_{01c}(f^o) = R^{A^d} - R_o^{A^d} + R^{B^d} - R_o^{B^d} \tag{75}$$

$$R_{01c}(f^s) - R_{01c}(f^o) = \delta^{A^d} + \delta^{B^d} \tag{76}$$

$$\text{where } \delta^{A^d} \triangleq R^{A^d} - R_o^{A^d} \tag{77}$$

$$\delta^{B^d} \triangleq R^{B^d} - R_o^{B^d}. \tag{78}$$

In the following, we prove that

$$\delta^{A^d} \leq 0 \text{ and } \delta^{B^d} \leq 0. \tag{79}$$

which would imply that

$$R_{01c}(f^s) - R_{01c}(f^o) \leq 0 \quad \forall f \in \mathcal{H} \neq f^s \tag{80}$$

$$\implies R_{01c}(f^o) \geq R_{01c}(f^s) \forall f \in \mathcal{H} \neq f^s \tag{81}$$

$$\implies f^s = f^*. \tag{82}$$

**Showing that $\delta^{A^d} \leq 0$ and $\delta^{B^d} \leq 0$** We first consider $\delta^{A^d}$:

$$\delta^{A^d} = R^{A^d} - R_o^{A^d} \tag{83}$$

$$= \int_{x \in A^d} 1 - \max_y \eta_y(x) p(x) dx - R_o^{A^d} \text{ using } 68. \tag{84}$$

We can (once again) further partition the space based on $f^o$. We divide $A^d$ in two based on the decision function of $f^o$:

$$A^{dz} \triangleq \{x; x \in A^d \text{ and } f^o = f_2^o(x, z)\} \tag{85}$$

$$A^{dx} \triangleq \{x; x \in A^d \text{ and } f^o = f_1^o(x)\}. \tag{86}$$

Continuing our development of $\delta^{A^d}$:

$$\delta^{A^d} = \int_{x \in A^d} 1 - \max_y \eta_y(x)p(x) - R_o^{A^d} \tag{87}$$

$$= \int_{x \in A^{dx}} 1 - \max_y \eta_y(x)p(x)dx - R_o^{A^{dx}} + \int_{x \in A^{dz}} 1 - \max_y \eta_y(x)p(x)dx - R_o^{A^{dz}} \tag{88}$$

$$= \int_{x \in A^{dx}} \left(1 - \max_y \eta_y(x)p(x) - \int_z \sum_y [\ell_{01c}(f^o(x,z), y)]p(y|x,z)p(z|x)p(x)\right)dx \tag{89}$$

$$+ \int_{x \in A^{dz}} \left(1 - \max_y \eta_y(x)p(x) - \int_z \sum_y [\ell_{01c}(f^o(x,z), y)]p(y|x,z)p(z|x)p(x)\right)dx \text{ using } 74 \tag{90}$$

$$= \int_{x \in A^{dx}} \left(1 - \max_y \eta_y(x) - \int_z \sum_y [\ell_{01c}(f_1^o(x), y)]p(y|x,z)p(z|x)\right)p(x)dx \tag{91}$$

$$+ \int_{x \in A^{dz}} \left(1 - \max_y \eta_y(x) - \int_z \sum_y [\ell_{01c}(f_2^o(x,z), y)]p(y|x,z)p(z|x)\right)p(x)dzdx \tag{92}$$

by def. of $A^{dx}, A^{dz}$ (93)

$$= \int_{x \in A^{dx}} \left(1 - \max_y \eta_y(x) - \sum_y \left(\mathbb{1}[f_1^o(x) \neq y)]\right)\eta_y(x)\right)p(x)dx \text{ [*]} \tag{94}$$

$$+ \int_{x \in A^{dz}} \left(1 - \max_y \eta_y(x) - \int_z \sum_y \mathbb{1}[f_2^o(x,z) \neq y](\zeta_y(x,z) + c)p(z|x)\right)p(x)dzdx \text{ [**]}. \tag{95}$$

$$\tag{96}$$

At this stage we can focus on one term at the time, starting with the part of $A^d$ where $f^o$ is not using $z$ which is the integral over $A^{dx}$:

$$[*] = \int_{x \in A^{dx}} \left(1 - \max_y \eta_y(x) - \sum_y \left(\mathbb{1}[f_1^o(x) \neq y)]\right)\eta_y(x)\right)p(x)dx \tag{97}$$

$$= \int_{x \in A^{dx}} \left(- \max_y \eta_y(x) + \eta_{f_1^o(x)}(x)\right)p(x)dx \tag{98}$$

no matter what value $f_1^o(x)$ takes, $\max_y \eta_y(x) \geq \eta_{y'}$. Therefore: (99)

$$[*] \leq 0. \tag{100}$$

Going to the second term [**], when $f^0$ uses $z$. We start by restating the definition of the set $A$:

$$A = \{x; \max \eta(x) \geq \mathbb{E}_{p(z|x)}[\max_y \zeta_y(x,z)] - c\}. \tag{101}$$

$$[**] = \int_{x \in A^{dz}} \left(1 - \max_y \eta_y(x) - \int_z \sum_y \mathbb{1}[f_2^o(x,z) \neq y](\zeta_y(x,z) + c)p(z|x)dz\right)p(x)dx \tag{102}$$

$$= \int_{x \in A^{dz}} \left(1 - \max_y \eta_y(x) - \mathbb{E}_{p(z|x)}[1 - \zeta_{f_2^o(x,z)}(x,z)]\right)p(x)dx \tag{103}$$

$$\leq \int_{x \in A^{dz}} \left(- \max_y \eta_y(x) + \mathbb{E}_{p(z|x)}[\max_y \zeta_y(x,z)] - c\right)p(x)dx \tag{104}$$

$$[**] \leq 0 \text{ by def. of } A. \tag{105}$$

Combining both results together, we have that

$$\delta^{A^d} = [*] + [**] \leq 0. \tag{106}$$

Following similar steps, we also have that

$$\delta^{B^d} \leq 0. \tag{107}$$

**Final step** Since we have both that $\delta^{B^d} \leq 0$ and $\delta^{A^d} \leq 0$, we can conclude that

$$R_{01c}(f^s) - R_{01c}(f^o) = \delta^{A^d} + \delta^{B^d} \tag{108}$$

$$R_{01c}(f^s) \leq R_{01c}(f^o) \,\forall f^o \neq f^s \tag{109}$$

$$\implies f^s = f^* \tag{110}$$

This concludes the proof. $\quad\square$

$\square$

## A.5 Proof of consistency for the multi class surrogate hinge loss

We start by restating our surrogate loss:

$$\ell_{hinge}^c(\mathbf{t}, \mathbf{v}, \tilde{r}, x, z, y) = (1 - \tilde{r}(x)) \sum_{y' \neq y} [\mathbf{t}_{y'}(x) + \frac{1}{K-1}]_+ \tag{111}$$

$$+ \tilde{r}(x) \left( \sum_{y' \neq y} [\mathbf{v}_{y'}(x, z) + \frac{1}{K-1}]_+ + \frac{Kc}{K-1} \right), \tag{112}$$

where we have $\mathbf{t} \in \mathbb{R}^K$ and $\mathbf{v} \in \mathbb{R}^K$ as the real-vectored outputs for $f_1$ and $f_2$ respectively with the constraints that $||\mathbf{v}||_1 = 0$ and $||\mathbf{t}||_1 = 0$, and $\tilde{r} \in [0, 1]$ as a soft decision output.

Since our surrogate optimization provides us with the triplet $\mathbf{t}, \mathbf{v}, \tilde{r}$, we map these to a two-stage classifier $f \in \mathcal{H}$ using the following a link function $\varphi : \mathbb{R}^K \times \mathbb{R}^K \times [0, 1] \to \mathcal{Y}$:

$$f = \varphi(\mathbf{t}, \mathbf{v}, r) = \begin{cases} \arg\max_{y \in \mathcal{Y}} \mathbf{t}_y(x) & \text{if } \tilde{r} < 0.5 \\ \arg\max_{y \in \mathcal{Y}} \mathbf{v}_y(x, z) & \text{o.w.} \end{cases} \in \mathcal{H}, \tag{113}$$

$$\text{where } f_1(x) = \arg\max_{y \in \mathcal{Y}} \mathbf{t}_y(x), \tag{114}$$

$$f_2(x, z) = \arg\max_{y \in \mathcal{Y}} \mathbf{v}_y(x, z), \tag{115}$$

$$r(x) = \mathbb{1}[\tilde{r}(x) < 0.5]. \tag{116}$$

We can then define our risk as usual:

$$R_{hinge}(\mathbf{t}, \mathbf{v}, \tilde{r}) = \mathbb{E}_{p(x,z,y)}[\ell_{hinge}^c(\mathbf{t}, \mathbf{v}, \tilde{r}, x, z, y)] \tag{117}$$

and consider the triplet of minimizers $\mathbf{t}^*(x), \mathbf{v}^*(x, z), \tilde{r}^*(x)$ of such a risk, which correspond to a two-stage solution:

$$\mathbf{t}^*, \mathbf{v}^*, \tilde{r}^* = \arg\min_{\mathbf{v}, \mathbf{t} \in \mathbb{R}^K, \tilde{r} \in [0,1]} R_{hinge}(\mathbf{t}, \mathbf{v}, \tilde{r}). \tag{118}$$

$$f_{hinge}^* = \varphi(\mathbf{t}^*, \mathbf{v}^*, \tilde{r}^*) \in \mathcal{H}. \tag{119}$$

We prove that our surrogate loss is Bayes-consistent w.r.t to the $\ell_{01c}$ loss by showing that 1)

$$f^* = f_{hinge}^*, \tag{120}$$

and 2)

$$R_{01c}(\varphi(\mathbf{t}, \mathbf{v}, \tilde{r})) - R_{01c}^* \leq \Psi\left(R_{hinge}(\mathbf{t}, \mathbf{v}, \tilde{r}) - R_{hinge}^*\right). \tag{121}$$

Taken together, those results guarantee that for any distribution $p(x, z, y)$, we have that:

$$R_{hinge}(\mathbf{t}, \mathbf{v}, \tilde{r}) \to R_{hinge}^* \implies R_{01c}(\varphi(\mathbf{t}, \mathbf{v}, \tilde{r})) \to R_{01c}^*. \tag{122}$$

which defines Bayes-consistency [Steinwart, 2007].

### A.5.1 The solutions $f^*$ of $R_{01c}(f)$ and $f^*_{hinge}$ coincide

Restating the definitions of the solution of the target and surrogate problems;

$$f^* = \arg\min_{f \in \mathcal{H}} R_{01c}(f) \tag{123}$$

$$f^*_{hinge} = \varphi(\mathbf{t}^*, \mathbf{v}^*, \tilde{r}^*) \tag{124}$$

$$\mathbf{t}^*, \mathbf{v}^*, \tilde{r}^* = \arg\min_{\mathbf{t}, \mathbf{v} \in \mathbb{R}^K, \tilde{r} \in [0,1]} R_{hinge}(\mathbf{t}, \mathbf{v}, \tilde{r}). \tag{125}$$

In this section, we show that the two solutions $f^*, f^*_{hinge}$ coincide.

**Lemma A.1.** *For any distribution $p(X, Z, Y)$;*

$$f^* = f^*_{hinge}. \tag{126}$$

*Proof.* We have previously shown in Appendix A.4 that the solution of our targeted problem

$$f^* = \arg\min_{f \in \mathcal{H}} R_{01c}(f) \tag{127}$$

is given by:

$$f^* = \begin{cases} f_1^*(x), & r^*(x) = 0 \\ f_2^*(x, z), & r(x) = 1 \end{cases}, \tag{128}$$

$$\text{where } f_1^*(x) = \arg\max_{y \in \mathcal{Y}} \eta_y(x), \quad \forall x \text{ s.t. } r^*(x) = 0 \tag{129}$$

$$f_2^*(x, z) = \arg\max_{y \in \mathcal{Y}} \zeta_y(x, z), \quad \forall x \text{ s.t. } r^*(x) = 1 \tag{130}$$

$$r^*(x) = \mathbb{1}\left[ max_y \eta_y(x) \le \mathbb{E}_{p(Z|x)}[\max_y \zeta_y(x, Z)] - c \right]. \tag{131}$$

Next, we show that the two-stage classifier obtained from the triplet minimizer of our surrogate loss $f^*_{hinge} = \varphi(\mathbf{t}^*, \mathbf{v}^*, \tilde{r}^*)$ corresponds to this solution.

$$\mathbf{t}^*, \mathbf{v}^*, \tilde{r}^* = \arg\min_{\mathbf{t}, \mathbf{v} \in \mathbb{R}^K, \tilde{r} \in [0,1]} R_{hinge}(\mathbf{t}, \mathbf{v}, \tilde{r}) \tag{132}$$

$$= \arg\min_{\mathbf{t}, \mathbf{v} \in \mathbb{R}^K, \tilde{r} \in [0,1]} \mathbb{E}_{p(x,z,y)}[\ell^c_{hinge}(\mathbf{t}, \mathbf{v}, \tilde{r}, x, z, y)] \tag{133}$$

$$\mathbf{t}^*, \mathbf{v}^*, \tilde{r}^* = \arg\min_{\mathbf{t}, \mathbf{v} \in \mathbb{R}^K, \tilde{r} \in [0,1]} \mathbb{E}_{p(x,z,y)}[(1 - \tilde{r}(x)) \sum_{y' \neq y} [\mathbf{t}_{y'} + \frac{1}{K-1}]_+ \tag{134}$$

$$+ \tilde{r}(x) \left( \sum_{y' \neq y} [\mathbf{v}_{y'} + \frac{1}{K-1}]_+ + \frac{Kc}{K-1} \right)]. \tag{135}$$

We can push the optimization problem inside the expectation w.r.t $p(x)$ as $\mathbf{t}(x), \mathbf{v}(x, z), \tilde{r}(x)$ are all functions of $x$ (and the inner expectation term is guaranteed to be bounded):

$$\mathbf{t}^*, \mathbf{v}^*, \tilde{r}^* = \arg\min_{\mathbf{t}, \mathbf{v} \in \mathbb{R}^K, \tilde{r} \in [0,1]} \mathbb{E}_{p(z,y|x)}[(1 - \tilde{r}(x)) \sum_{y' \neq y} [\mathbf{t}_{y'} + \frac{1}{K-1}]_+ \tag{136}$$

$$+ \tilde{r}(x) \left( \sum_{y' \neq y} [\mathbf{v}_{y'} + \frac{1}{K-1}]_+ + \frac{Kc}{K-1} \right)]. \tag{137}$$

Since the loss is a linear combination of two terms that respectively depend on $\mathbf{t}$ and $\mathbf{v}$, we can see that for any $\tilde{r}$, the minimizers for $\mathbf{t}$ and $\mathbf{v}$ will always be equal to the minimizer of the individual terms:

$$\mathbf{t}^*(x) = \arg\min_{\mathbf{t} \in \mathbb{R}^K} \mathbb{E}_{p(y|x)}[\sum_{y' \neq y} [\mathbf{t}_{y'} + \frac{1}{K-1}]_+] \text{ (for any } \mathbf{v}, \tilde{r}(x) < 1) \tag{138}$$

$$\mathbf{v}^*(x, z) = \arg\min_{\mathbf{v} \in \mathbb{R}^K} \mathbb{E}_{p(y|x,z)}[\sum_{y' \neq y} [\mathbf{v}_{y'} + \frac{1}{K-1}]_+ + \frac{Kc}{K-1}] \text{ (for any } \mathbf{t}, \tilde{r}(x) \ge 0). \tag{139}$$

For the multi class hinge loss that we are considering, it is known that the minimizing functions are given by the following [Tarigan and van de Geer, 2008]:

$$\mathbf{t}_y^*(x) = \begin{cases} 1 & \text{if } y = \arg\max_{y \in \mathcal{Y}} \eta_y(x) \\ \frac{-1}{K-1} & \text{o.w.} \end{cases} \tag{140}$$

$$\mathbf{v}_y^*(x, z) = \begin{cases} 1 & \text{if } y = \arg\max_{y \in \mathcal{Y}} \zeta_y(x, z) \\ \frac{-1}{K-1} & \text{o.w.} \end{cases} \tag{141}$$

which gets converted into $f_1^*$ and $f_2^*$ by the link function $\varphi$ (see Eqn 10):

$$f_{hinge,1}(x) = \arg\max_{y \in \mathcal{Y}} \mathbf{t}_y^*(x) \quad \forall x \text{ s.t. } \tilde{r}(x) < 1 \tag{142}$$

$$= \arg\max_{y \in \mathcal{Y}} \eta_y(x) \quad \forall x \text{ s.t. } \tilde{r}(x) < 1 \text{ by Eqn. 140}, \tag{143}$$

$$f_{hinge,1}(x) = f_1^*(x) \quad \forall x \text{ s.t. } \tilde{r}(x) < 1 \text{ by Eqn. 54}, \tag{144}$$

$$f_{hinge,2}(x) = f_2^*(x) \quad \forall x \text{ s.t. } \tilde{r}(x) \geq 0 \text{ by Eqn. 55}. \tag{145}$$

Next, we turn to the decision function $\tilde{r}(x)$. Using $\tilde{r}$ as shorthands for $\tilde{r}(x)$:

$$\tilde{r}^* = \arg\min_{\tilde{r} \in [0,1]} \mathbb{E}_{p(z,y|x)}[\ell_{hinge}^c(x, z, y, \mathbf{t}^*, \mathbf{v}^*, \tilde{r})] \tag{146}$$

$$= \arg\min_{\tilde{r} \in [0,1]} (1 - \tilde{r}(x)) \mathbb{E}_{p(y|x)}[\sum_{y' \neq y} [\mathbf{t}_{y'}^* + \frac{1}{K-1}]_+] \tag{147}$$

$$+ \tilde{r}(x) \mathbb{E}_{p(y,z|x)}\left[ \left( \sum_{y' \neq y} [\mathbf{v}_{y'}^* + \frac{1}{K-1}]_+ + \frac{Kc}{K-1} \right) \right]. \tag{148}$$

Since it is a linear combination of $1 - \tilde{r}(x)$ and $\tilde{r}(x)$, it is clear that the minimizer $\tilde{r}^*(x)$ will either be at 0 or 1. We can therefore rewrite the optimization problem as the following:

$$A(x, \tilde{r}) = (1 - \tilde{r}(x)) \mathbb{E}_{p(y|x)}[\sum_{y' \neq y} [\mathbf{t}_{y'}^* + \frac{1}{K-1}]_+] \tag{149}$$

$$+ \tilde{r}(x) \mathbb{E}_{p(y,z|x)}[\frac{Kc}{K-1} + \left( \sum_{y' \neq y} +[\mathbf{v}_{y'}^* + \frac{1}{K-1}]_+ \right)] \tag{150}$$

$$= (1 - \tilde{r}(x)) \sum_{y \in \mathcal{Y}} \eta_y(x, z) \sum_{y' \neq y} [\mathbf{t}_{y'}^* + \frac{1}{K-1}]_+ \tag{151}$$

$$+ \tilde{r}(x) \left( \frac{Kc}{K-1} + \mathbb{E}_{p(z|x)} \left[ \sum_{y \in \mathcal{Y}} \zeta_y(x, z) \sum_{y' \neq y} [\mathbf{v}_{y'}^* + \frac{1}{K-1}]_+ \right] \right) \tag{152}$$

$$\tilde{r}^*(x) = \arg\min_{\tilde{r} \in \{0,1\}} A(x, \tilde{r}). \tag{153}$$

We consider the two cases for $A(x, \tilde{r})$.

$$A(x, 0) = \sum_{y \in \mathcal{Y}} \eta_y(x) \sum_{y' \neq y} [\mathbf{t}_{y'}^* + \frac{1}{K-1}]_+ \tag{154}$$

$$= \sum_{y \neq \arg\max_y \eta_y(x)} \eta_y(x) \sum_{y' \neq y} [\mathbf{t}_{y'}^* + \frac{1}{K-1}]_+ \tag{155}$$

$$+ \eta_{\arg\max_y \eta_y(x)}(x) \sum_{y' \neq \arg\max_y \eta_y(x)} [\mathbf{t}_{y'}^* + \frac{1}{K-1}]_+ \tag{156}$$

$$= \sum_{y \neq \arg\max_y \eta_y(x)} \eta_y(x) \left( (K-2)[\frac{-1}{K-1} + \frac{1}{K-1}]_+ + [1 + \frac{1}{K-1}]_+ \right) \tag{157}$$

$$+ \eta_{\arg\max_y \eta_y(x)}(x) \sum_{y' \neq \arg\max_y \eta_y(x)} [\frac{-1}{K-1} + \frac{1}{K-1}]_+ \text{ by def of } \mathbf{t}^* \text{ Eqn.140} \tag{158}$$

$$A(x, 0) = \frac{K}{K-1} \left( 1 - \max_{y \in \mathcal{Y}} \eta_y(x) \right). \tag{159}$$

For the second case, using similar steps:

$$A(x, 1) = \frac{Kc}{K-1} + \mathbb{E}_{p(z|x)} \left[ \sum_{y \in \mathcal{Y}} \zeta_y(x, z) \sum_{y' \neq y} [\mathbf{v}_{y'}^* + \frac{1}{K-1}]_+ \right] \tag{160}$$

$$= \frac{K}{K-1} (\mathbb{E}_{p(z|x)} [ \left( 1 - \max_{y \in \mathcal{Y}} \zeta_y(x) \right) ] + c). \tag{161}$$

This allows us to write the solution as

$$\tilde{r}^*(x) = \begin{cases} 0 & \text{if } A(r=0, x) < A(r=1, x) \\ 1 & \text{o.w.} \end{cases} \tag{162}$$

$$= \begin{cases} 0 & \text{if } \frac{K}{K-1} (1 - \max_{y \in \mathcal{Y}} \eta_y(x)) \leq \frac{K}{K-1} (\mathbb{E}_{p(z|x)} [(1 - \max_{y \in \mathcal{Y}} \zeta_y(x))] + c) \\ 1 & \text{o.w.} \end{cases} \tag{163}$$

$$\tilde{r}^*(x) = \begin{cases} 0 & \text{if } \max_y \eta_y(x) \geq \mathbb{E}_{p(z|x)} [ \max \zeta_y(x, z) ] - c \\ 1 & \text{o.w.} \end{cases} \tag{164}$$

We have therefore shown that

$$\tilde{r}^*(x) = r^*(x). \tag{165}$$

Since we now have that $\tilde{r}^*(x)$ is restricted to the binary values $\tilde{r}^*(x) = \{0, 1\}$, we can rewrite the optimal classifiers that we previously obtained:

$$f_{hinge,1}^*(x) = f_1^*(x) \quad \forall x \text{ s.t. } \tilde{r}(x) < 1 \tag{166}$$

$$f_{hinge,2}^*(x) = f_2^*(x) \quad \forall x \text{ s.t. } \tilde{r}(x) \geq 0 \tag{167}$$

as

$$f_{hinge,1}^*(x) = f_1^*(x) \quad \forall x \text{ s.t. } r^*(x) = 0, \tag{168}$$

$$f_{hinge,2}^*(x) = f_2^*(x) \quad \forall x \text{ s.t. } r^*(x) = 1. \tag{169}$$

$$f_{hinge,1}^*(x) = f_1^*(x), \tag{170}$$

$$f_{hinge,2}^*(x) = f_2^*(x). \tag{171}$$

Therefore, we can see that the optimal $\mathbf{t}^*$ and $\mathbf{v}^*$ leads to the same solution for the internal classifiers of $f^*$. We have therefore shown that:

$$f_{hinge,1}^*(x) = f_1^*(x), \tag{172}$$

$$f_{hinge,2}^*(x) = f_2^*(x), \tag{173}$$

$$\text{and } \tilde{r}^*(x) = r^*(x), \tag{174}$$

$$\implies f^* = f_{hinge}^*. \tag{175}$$

This concludes the proof. □

### A.5.2 Gap of the hinge loss

Next, we aim to show that for some increasing function $\Psi$ with $\Psi(0) = 0$, we can upper bound the risk gap of our loss of interest $R_{01c}(\varphi(\mathbf{t}, \mathbf{v}, \tilde{r})) - R_{01c}^*$ with the risk gap of our surrogate hinge loss $R_{hinge}(\mathbf{t}, \mathbf{v}, \tilde{r}) - R_{hinge}^*$.

**Lemma A.2.** *For any distribution $p(X, Z, Y)$:*

$$R_{01c}(\varphi(\mathbf{t}, \mathbf{v}, \tilde{r})) - R_{01c}^* \leq \Psi\left(R_{hinge}(\mathbf{t}, \mathbf{v}, \tilde{r}) - R_{hinge}^*\right). \tag{176}$$

*Proof.* We start by developing the hinge risk $R_{hinge}(\mathbf{t}, \mathbf{v}, \tilde{r})$:

$$R_{hinge}(\mathbf{t}, \mathbf{v}, \tilde{r}) = \mathbb{E}_{p(x,z,y)}[\ell_{hinge}^c(\mathbf{t}, \mathbf{v}, \tilde{r}, x, z, y)] \tag{177}$$

$$= \mathbb{E}_{p(x,z)}[\sum_{y \in \mathcal{Y}} \zeta_y(x, z)(1 - \tilde{r}(x)) \sum_{y' \neq y} [\mathbf{t}_{y'} + \frac{1}{K-1}]_+ \tag{178}$$

$$+ \sum_{y \in \mathcal{Y}} \zeta_y(x, z)\tilde{r}(x) \left(\sum_{y' \neq y} [\mathbf{v}_{y'} + \frac{1}{K-1}]_+ + \frac{Kc}{K-1}\right)]. \tag{179}$$

Next we develop the term associated to the optimal hinge risk $R_{hinge}^*$:

$$R_{hinge}^* = \mathbb{E}_{p(x,z)}[\sum_{y \in \mathcal{Y}} \zeta_y(x, z)(1 - \tilde{r}^*(x)) \sum_{y' \neq y} [\mathbf{t}_{y'}^* + \frac{1}{K-1}]_+ \tag{180}$$

$$+ \sum_{y \in \mathcal{Y}} \zeta_y(x, z)\tilde{r}^*(x) \left(\sum_{y' \neq y} [\mathbf{v}_{y'}^* + \frac{1}{K-1}]_+ + \frac{Kc}{K-1}\right)]. \tag{181}$$

$$= \mathbb{E}_{p(x)}[\mathbb{1}[\tilde{r}^*(x) = 0]A(x, 0)] + \mathbb{E}_{p(x,z)}[\mathbb{1}[\tilde{r}^*(x) = 1]A(x, 1)] \text{ reusing Eqn 161, 159.} \tag{182}$$

$$R_{hinge}^* = \frac{K}{K-1}\mathbb{E}_{p(x)}[\mathbb{1}[\tilde{r}^*(x) = 0]1 - \max_{y \in \mathcal{Y}} \eta_y(x)] \tag{183}$$

$$+ \frac{K}{K-1}\mathbb{E}_{p(x,z)}[\mathbb{1}[\tilde{r}^*(x) = 1](1 - \max_{y \in \mathcal{Y}} \zeta_y(x) + c)]. \tag{184}$$

Bringing both $R_{hinge}^*$ and $R_{hinge}(f)$ to evaluate the gap $G \triangleq R_{hinge}(f) - R_{hinge}^*$, we can decompose the gap by a sum of 4 terms that are driven by the ground truth decision cases, i.e. $r^* = 0$ or $^* = 1$ and the decision of the model $\tilde{r}(x)$:

$$G \triangleq R_{hinge}(\mathbf{t}, \mathbf{v}, \tilde{r}) - R_{hinge}^* \tag{185}$$

$$G = \mathbb{E}_{p(x,z)}[\sum_{y \in \mathcal{Y}} \zeta_y(x, z)(1 - \tilde{r}(x)) \sum_{y' \neq y} [\mathbf{t}_{y'} + \frac{1}{K-1}]_+ \tag{186}$$

$$+ \sum_{y \in \mathcal{Y}} \zeta_y(x, z)\tilde{r}(x) \left(\sum_{y' \neq y} [\mathbf{v}_{y'} + \frac{1}{K-1}]_+ + \frac{Kc}{K-1}\right)] \tag{187}$$

$$- \frac{K}{K-1}\mathbb{E}_{p(x)}[\mathbb{1}[\tilde{r}^*(x) = 0](1 - \max_{y \in \mathcal{Y}} \eta_y(x))] + \mathbb{E}_{p(x,z)}[\mathbb{1}[\tilde{r}^*(x) = 1](c + 1 - \max_{y \in \mathcal{Y}} \zeta_y(x, z))]. \tag{188}$$

We can define the corresponding gap to each case as follows:

$$G_1 \triangleq \mathbb{1}[\tilde{r}^*(x) = 0]\mathbb{1}[\tilde{r}(x) \leq 0.5] \sum_{y \in \mathcal{Y}} \eta_y(x,z)(1 - \tilde{r}(x)) \sum_{y' \neq y}[\mathbf{t}_{y'} + \frac{1}{K-1}]_+ \tag{189}$$

$$+ \sum_{y \in \mathcal{Y}} \zeta_y(x,z)\tilde{r}(x) \left( \sum_{y' \neq y}[\mathbf{v}_{y'} + \frac{1}{K-1}]_+ + \frac{Kc}{K-1} \right) - \frac{K}{K-1}(1 - \max_{y \in \mathcal{Y}} \eta_y(x)) \tag{190}$$

$$G_2 \triangleq \mathbb{1}[\tilde{r}^*(x) = 0]\mathbb{1}[\tilde{r}(x) \geq 0.5] \sum_{y \in \mathcal{Y}} \eta_y(x,z)(1 - \tilde{r}(x)) \sum_{y' \neq y}[\mathbf{t}_{y'} + \frac{1}{K-1}]_+ \tag{191}$$

$$+ \sum_{y \in \mathcal{Y}} \zeta_y(x,z)\tilde{r}(x) \left( \sum_{y' \neq y}[\mathbf{v}_{y'} + \frac{1}{K-1}]_+ + \frac{Kc}{K-1} \right) - \frac{K}{K-1}(1 - \max_{y \in \mathcal{Y}} \eta_y(x))$$
$$\tag{192}$$

$$G_3 \triangleq \mathbb{1}[\tilde{r}^*(x) = 1]\mathbb{1}[\tilde{r}(x) \leq 0.5] \sum_{y \in \mathcal{Y}} \eta_y(x,z)(1 - \tilde{r}(x)) \sum_{y' \neq y}[\mathbf{t}_{y'} + \frac{1}{K-1}]_+ \tag{193}$$

$$+ \sum_{y \in \mathcal{Y}} \zeta_y(x,z)\tilde{r}(x) \left( \sum_{y' \neq y}[\mathbf{v}_{y'} + \frac{1}{K-1}]_+ + \frac{Kc}{K-1} \right) - \frac{K}{K-1}(c + 1 - \max_{y \in \mathcal{Y}} \zeta_y(x,z)),$$
$$\tag{194}$$

$$G_4 \triangleq \mathbb{1}[\tilde{r}^*(x) = 1]\mathbb{1}[\tilde{r}(x) \geq 0.5] \sum_{y \in \mathcal{Y}} \eta_y(x,z)(1 - \tilde{r}(x)) \sum_{y' \neq y}[\mathbf{t}_{y'} + \frac{1}{K-1}]_+ \tag{195}$$

$$+ \sum_{y \in \mathcal{Y}} \zeta_y(x,z)\tilde{r}(x) \left( \sum_{y' \neq y}[\mathbf{v}_{y'} + \frac{1}{K-1}]_+ + \frac{Kc}{K-1} \right) - \frac{K}{K-1}(c + 1 - \max_{y \in \mathcal{Y}} \zeta_y(x,z)),$$
$$\tag{196}$$

and rewrite the total gap as:

$$G = \mathbb{E}_{p(x,z)}\big[G_1 + G_2 + G_3 + G_4\big]. \tag{197}$$

We can obtain a similar decomposition for the risk gap of our target risk $R_{01c}(f)$. We recall the definition:

$$R_{01c}(f) = \mathbb{E}_{p(x,z,y)}\Big[\ell_{01c}(f(x,z),y)\Big] \tag{198}$$

$$= \mathbb{E}_{p(x,z,y)}\Big[\mathbb{1}[r(x) = 0]\mathbb{1}[f_1(x) \neq y] + \mathbb{1}[r(x) = 1][\mathbb{1}[f_2(x,z) \neq y)] + c]\Big] \tag{199}$$

$$= \mathbb{E}_{p(x)}\Big[\mathbb{1}[r(x) = 0]\mathbb{E}_{p(y|x)}\mathbb{1}[f_1(x) \neq y] + \mathbb{1}[r(x) = 1]\mathbb{E}_{p(y,z|x)}[\mathbb{1}[f_2(x,z) \neq y)] + c]\Big]$$
$$\tag{200}$$

$$R_{01c}(f) = \mathbb{E}_{p(x)}\Big[\mathbb{1}[r(x) = 0](1 - \eta_{f_1(x)}) + \mathbb{1}[r(x) = 1]\mathbb{E}_{p(z|x)}[1 - \zeta_{f_2(x,z)} + c]\Big] \tag{201}$$

and decompose the gap risk with terms based on similar cases:

$$F_1 \triangleq \mathbb{1}[r(x) = 0, r^*(x) = 0](\eta_{f^{*1}(x)} - \eta_{f_1(x)}) \tag{202}$$

$$F_2 \triangleq \mathbb{1}[r(x) = 0, r^*(x) = 1](\mathbb{E}_{p(z|x)}[\zeta_{f_2^*(x,z)}] - \eta_{f_1(x)} - c) \tag{203}$$

$$F_3 \triangleq \mathbb{1}[r(x) = 1, r^*(x) = 0](\eta_{f^{*1}(x)} - \mathbb{E}_{p(z|x)}[\zeta_{f_2(x,z)}] + c) \tag{204}$$

$$F_4 \triangleq \mathbb{1}[r(x) = 1, r^*(x) = 1](\mathbb{E}_{p(z|x)}[\zeta_{f_2^*(x,z)} - \zeta_{f_2(x,z)}]) \tag{205}$$

$$R_{01c}(f) - R_{01c}^* = \mathbb{E}_{p(x,z)}\Big[F_1 + F_2 + F_3 + F_4\Big]. \tag{206}$$

This makes sense: if the optimal decision is defer and we don't, the risk is diminished by the saved computation $c$ (the second case). If the optimal decision is not to defer and do, the risk is increased by the computation cost $c$ (the third case).

To prove the result, we show that :

$$\mathbb{E}_{p(x,z)}\Big[F_1\Big] \leq \Psi(\mathbb{E}_{p(x,z)}[G_1]) \tag{207}$$

$$\mathbb{E}_{p(x,z)}\Big[F_2\Big] \leq \Psi(\mathbb{E}_{p(x,z)}[G_2]) \tag{208}$$

$$\mathbb{E}_{p(x,z)}\Big[F_3\Big] \leq \Psi(\mathbb{E}_{p(x,z)}[G_3]) \tag{209}$$

$$\mathbb{E}_{p(x,z)}\Big[F_4\Big] \leq \Psi(\mathbb{E}_{p(x,z)}[G_4]). \tag{210}$$

$G_1$ **inequality**    Starting with $G_1$ and $F_1$:

$$G_1 = \mathbb{1}[\tilde{r}^*(x) = 0]\mathbb{1}[\tilde{r}(x) \leq 0.5]\sum_{y \in \mathcal{Y}} \eta_y(x, z)(1 - \tilde{r}(x))\sum_{y' \neq y}[\mathbf{t}_{y'} + \frac{1}{K-1}]_+ \tag{211}$$

$$+ \sum_{y \in \mathcal{Y}} \zeta_y(x, z)\tilde{r}(x)\left(\sum_{y' \neq y}[\mathbf{v}_{y'} + \frac{1}{K-1}]_+ + \frac{Kc}{K-1}\right) - \frac{K}{K-1}(1 - \max_{y \in \mathcal{Y}}\eta_y(x)) \tag{212}$$

$$\geq \frac{K}{K-1}(1 - \tilde{r}(x))(1 - \eta_{f1}) + \tilde{r}(x)(1 - \zeta_{f2} + c)] - (1 - \eta_*) \text{ by def. of the hinge loss} \tag{213}$$

$$= \frac{K}{K-1}(1 - \eta_{f1} - \tilde{r}(x) + \tilde{r}(x)\eta_{f1} + \tilde{r}(x) - \tilde{r}(x)\zeta_{f2} + \tilde{r}(x)c - 1 + \eta_*) \tag{214}$$

$$G_1 \geq \mathbb{1}[\tilde{r}(x) \leq 0.5]\frac{K}{K-1}\left(\eta_* - \eta_{f1} + \tilde{r}(x)(\eta_{f1} - \zeta_{f2} + c)\right). \tag{215}$$

We need to find a mapping $\Psi$ such that the following holds:

$$\mathbb{E}_{p(x,z)}\Big[F_1\Big] \leq \Psi(\mathbb{E}_{p(x,z)}[G_1]) \tag{216}$$

$$\mathbb{E}_{p(x,z)}\Big[(\eta_{f^{*1}} - \eta_{f_1})\Big] \leq \Psi(\mathbb{E}_{p(x,z)}[G_1]). \tag{217}$$

Using the simple scaling function $\Psi(x) = \frac{2(K-1)}{K}x$, we can see that the previous inequality holds:

$$\mathbb{E}_{p(x,z)}[\frac{2(K-1)G_1}{K}] \geq 2\mathbb{E}_{p(x,z)}\Big[\mathbb{1}[\tilde{r}(x) \leq 0.5](\eta_* - \eta_{f1} + \tilde{r}(x)(\eta_{f1} - \zeta_{f2} + c))\Big] \tag{218}$$

$$\geq 2\mathbb{E}_{p(x,z)}\Big[\mathbb{1}[\tilde{r}(x) \leq 0.5](\eta_* - \eta_{f1} + \tilde{r}(x)(\eta_{f1} - \zeta_* + c))\Big] \tag{219}$$

$$\geq 2\mathbb{E}_{p(x,z)}\Big[\mathbb{1}[\tilde{r}(x) \leq 0.5](\eta_* - \eta_{f1} + \tilde{r}(x)(\eta_{f1} - (c + \eta_*) + c))\Big] \tag{220}$$

$$\text{as } r^* = 0 \tag{221}$$

$$\geq 2\mathbb{E}_{p(x,z)}\Big[\mathbb{1}[\tilde{r}(x) \leq 0.5](\eta_* - \eta_{f1} - \tilde{r}(x)(\eta_* - \eta_{f1}))\Big] \tag{222}$$

$$\geq 2\mathbb{E}_{p(x,z)}\Big[\mathbb{1}[\tilde{r}(x) \leq 0.5](1 - \tilde{r}(x))(\eta_* - \eta_{f1})\Big] \tag{223}$$

$$\geq 2\mathbb{E}_{p(x,z)}\Big[0.5(\eta_* - \eta_{f1})\Big] \tag{224}$$

$$\Psi(\mathbb{E}_{p(x,z)}[G_1]) \geq \mathbb{E}_{p(x,z)}\Big[(\eta_* - \eta_{f1})\Big] \text{ with } \Psi(x) = \frac{2(K-1)}{K}x. \quad \square \tag{225}$$

$G_2$ **inequality** For the next inequality with $G_2$ and $F_2$, again using the same function $\Psi(x) = \frac{2(K-1)}{K}x$, the inequality holds:

$$\mathbb{E}_{p(x,z)}[\frac{2(K-1)G_2}{K}] \geq 2\mathbb{E}_{p(x,z)}\Big[\mathbb{1}[\tilde{r}(x) \geq 0.5](\eta_* + \eta_{f1}(\tilde{r}(x) - 1) - \tilde{r}(x)\zeta_{f2} + \tilde{r}(x)c)\Big] \quad (226)$$

$$\geq 2\mathbb{E}_{p(x,z)}\Big[\mathbb{1}[\tilde{r}(x) \geq 0.5](\eta_* + \eta_*(\tilde{r}(x) - 1) - \tilde{r}(x)\zeta_{f2} + \tilde{r}(x)c)\Big] \quad (227)$$

$$= 2\mathbb{E}_{p(x,z)}\Big[\mathbb{1}[\tilde{r}(x) \geq 0.5](\tilde{r}(x)(\eta_* - \zeta_{f2} + c)\Big] \quad (228)$$

$$\geq 2\mathbb{E}_{p(x,z)}\Big[0.5(\eta_* - \zeta_{f2} + c)\Big] \quad (229)$$

$$\Psi(\mathbb{E}_{p(x,z)}[G_2]) \geq \mathbb{E}_{p(x,z)}\Big[F_2\Big] \quad \Box. \quad (230)$$

$$(231)$$

Following similar steps, we obtain

$$\Psi(\mathbb{E}_{p(x,z)}[G_3]) \geq \mathbb{E}_{p(x,z)}\Big[F_3\Big] \quad (232)$$

$$\Psi(\mathbb{E}_{p(x,z)}[G_4]) \geq \mathbb{E}_{p(x,z)}\Big[F_4\Big]. \quad (233)$$

Putting all results together, we obtain

$$R_{01c}(\varphi(\mathbf{t}, \mathbf{v}, \tilde{r})) - R_{01c}^* \leq \frac{2(K-1)}{K}\Big(R_{hinge}(\mathbf{t}, \mathbf{v}, \tilde{r}) - R_{hinge}^*\Big) \quad (234)$$

$$R_{01c}(\varphi(\mathbf{t}, \mathbf{v}, \tilde{r})) - R_{01c}^* \leq \Psi\Big(R_{hinge}(\mathbf{t}, \mathbf{v}, \tilde{r}) - R_{hinge}^*\Big) \quad \Box. \quad (235)$$

with $\Psi(0) = 0$ and is increasing. This concludes the proof.

$\Box$

## A.6 Proof of the failure of the cross entropy version

In this section, we prove that the cross entropy version of the surrogate loss we presented cannot be Bayes-consistent.

We recall the entropy version, with $\mathbf{p}^1 \in \Delta^K$ for $f_1$ and $\mathbf{p}^2 \in \Delta^K$ for $f_2$, with the same link function $\varphi$. The cross entropy version of the loss that we consider is given by:

$$\ell_{ce}^{g(c)}(\mathbf{p}^1, \mathbf{p}^2, \tilde{r}, x, z, y) = -\Big((1 - \tilde{r}(x))\log(\mathbf{p}_y^1) + \tilde{r}(x)(\log(\mathbf{p}_y^2) + g(c))\Big) , \quad (236)$$

where $g(c)$ is an arbitrary function. We again consider its associated risk:

$$R_{ce}(\mathbf{p}^1, \mathbf{p}^2, \tilde{r}) = \mathbb{E}_{p(x,z,y)}[\ell_{ce}^{g(c)}(\mathbf{p}^1, \mathbf{p}^2, \tilde{r}, x, z, y)] \quad (237)$$

and minimizing function:

$$f_{ce}^* = \underset{\mathbf{p}^1, \mathbf{p}^2, \tilde{r} \in [0,1]}{\arg\min} R_{ce}(\mathbf{p}^1, \mathbf{p}^2, \tilde{r}). \quad (238)$$

**Lemma 4.2.** Cross-entropy surrogate loss is not Bayes Consistent.
There is no $g(c)$ for which:

$$f^* = f_{ce}^*. \quad (239)$$

*Proof.* Following a similar reasoning as the proof of Lemma A.1 , we can readily find that the optimal predicted probability vectors $\mathbf{p}^1$ and $\mathbf{p}^2$ in $f_{ce}^*$ should match the posteriors:

$$f_{ce}^* = \{\eta(x), \zeta(x, z), \tilde{r}_{ce}^*\}. \quad (240)$$

Now, to find the optimal decision function of the cross entropy risk $\tilde{r}_{ce}^*$, we can again obtain the solution as

$$\tilde{r}_{ce}^*(x) = \underset{\tilde{r} \in \{0,1\}}{\arg\min} B(\tilde{r}, x) \text{ where} \quad (241)$$

$$B(\tilde{r}, x) = \mathbb{E}_{p(x,z)}[-(1 - \tilde{r}(x))\log(\eta_*(x)) - \tilde{r}(x)(\log(\zeta_*(x, z)) + g(c))]. \quad (242)$$

by following the same steps that were taking to obtain Eqn. 153. We again consider the two cases:

$$B(0, x) = -\log(\eta_*(x)),\tag{243}$$

$$B(1, x) = -\mathbb{E}_{p(z|x)}[\log(\zeta_*(x, z))] - g(c).\tag{244}$$

The solution for the cross-entropy decision function is hence given by:

$$\tilde{r}^*_{ce}(x) = \begin{cases} 0 & \text{if } B(0, x) < B(1, x), \\ 1 & \text{o.w.} \end{cases}\tag{245}$$

$$\tilde{r}^*_{ce}(x) = \begin{cases} 0 & \text{if } \log(\eta_*(x)), \geq \mathbb{E}_{p(z|x)}[\log(\zeta_*(x, z))] + g(c) \\ 1 & \text{o.w.} \end{cases}\tag{246}$$

If we recall the solution decision of our target problem;

$$r^*(x) = \eta_*(x) < \mathbb{E}_{p(z|x)}[\zeta_*(x, z)] + c,\tag{247}$$

we are searching for a function $g(c)$ for which

$$\mathbb{1}[\log(\eta_*(x)) < \mathbb{E}_{p(z|x)}[\log(\zeta_*(x, z))] + g(c)] = r^*(x) \quad \forall x, \eta, \zeta.\tag{248}$$

If we define the decision sets of $x$:

$$\mathcal{D}_1 := \{x \mid r^*(x) = 1\},\tag{249}$$

$$\mathcal{D}_0 := \{x \mid r^*(x) = 0\},\tag{250}$$

we can rewrite the condition Eqn. 248 as:

$$\log(\eta_*(x)) - \mathbb{E}_{p(z|x)}\big[\log(\zeta_*(x, z))\big] < g(c) \quad \text{for all } x \in \mathcal{D}_1,\tag{251}$$

$$\log(\eta_*(x)) - \mathbb{E}_{p(z|x)}\big[\log(\zeta_*(x, z))\big] \geq g(c) \quad \text{for all } x \in \mathcal{D}_0.\tag{252}$$

This implies that the function $g(c)$ should satisfy:

$$\sup_{x \in \mathcal{D}_1} \log(\eta_*(x)) - \mathbb{E}_{p(z|x)}\big[\log(\zeta_*(x, z))\big] < g(c) \leq \inf_{x \in \mathcal{D}_0} \log(\eta_*(x)) - \mathbb{E}_{p(z|x)}\big[\log(\zeta_*(x, z))\big] \forall \eta, \zeta.\tag{253}$$

However, our condition for $r^*(x)$ is on the absolute scale, not on the log scale $r^*(x) = \eta_*(x) < \mathbb{E}_{p(z|x)}[\zeta_*(x, z)] + c$. We will therefore have that for some $\eta, \zeta$

$$\sup_{x \in \mathcal{D}_1} \log(\eta_*(x)) - \mathbb{E}_{p(z|x)}\big[\log(\zeta_*(x, z))\big] > \inf_{x \in \mathcal{D}_0} \log(\eta_*(x)) - \mathbb{E}_{p(z|x)}\big[\log(\zeta_*(x, z))\big].\tag{254}$$

This implies that there is no $g(c)$ that can satisfy

$$\mathbb{1}[\log(\eta_*(x)) < \mathbb{E}_{p(z|x)}[\log(\zeta_*(x, z))] + g(c)] = r^*(x).\tag{255}$$

This concludes the proof. $\qquad\square$

### A.7 Extension to the multi-classifier setting

In this section, we provide a high-level view of how we could extend our results to the more general case of $L + 1$ classifiers. In this setting, there are still two stages, but there are $L$ classifiers to choose from at the second stage (with associated costs $c_1, \ldots c_L$). We would need to introduce additional random variables $Z_1, \ldots, Z_L$, and the loss in Eqn 2 would need to be generalized to multiple classifiers and costs:

$$\ell_{01c}(f(x, z_{1:L}), y) = \begin{cases} \mathbb{1}[f_1(x) \neq y] & r(x) = 0 \\ \mathbb{1}[f_2(x, z_1) \neq y] + c_1 & r(x) = 1 \\ \ldots, \\ \mathbb{1}[f_L(x, z_L) \neq y] + c_L & r(x) = L. \end{cases}\tag{256}$$

The corresponding solution for the decision boundaries (Eqn. 9) would become more complex. Instead of comparing the maximum posterior probability $\eta(x)$ to a single expected future gain minus cost, the comparison would now need to be made against **the best** potential expected future gain for each classifier:

$$
r^*(x) = \begin{cases}
0 & \text{if } 1\Big[\max_y \eta_y(x) > \max_{l\in 1,L}\Big(\mathbb{E}_{p(Z_l|x)}[\max_y p(y|x, Z_l)] - c_l\Big)\Big] \\
1 & \text{if } 1 = \max_{l\in 1,L}\Big(\mathbb{E}_{p(Z_l|x)}[\max_y p(y|x, Z_l)] - c_l\Big) \\
\dots & \\
L & \text{if } L = \max_{l\in 1,L}\Big(\mathbb{E}_{p(Z_l|x)}[\max_y p(y|x, Z_l)] - c_l\Big).
\end{cases}
\tag{257}
$$

The term $\max_{l\in 1,L}\Big(\mathbb{E}_{p(Z_l|x)}[\max_y p(y|x, Z_l)] - c_l\Big)$ returns the index of the best model that can be used at the second stage.

Then, we could propose a multi-classifier surrogate loss (replacing Eqn. 11) by using a soft decision function that is now a multiclass probability vector $\tilde{r}(x) \in [0, 1]^{L+1}$ with $\sum_{l=0}^{L} \tilde{r}_l(x) = 1$. We would also need to introduce $L$ new learnable vectors $\mathbf{t}^{(0)}, \dots, \mathbf{t}^{(L)}$, and an index-dependent cost term $\frac{K}{K-1}g(l, c_1, \dots, c_L, K)$, where $g(l, c_1, \dots, c_L, K)$ is some linear function of the costs that would need to be derived and obtained from the proof. The hinge surrogate loss could (potentially) have the following form:

$$
\ell^c_{hinge}(\mathbf{t}^{(0)}, \dots, \mathbf{t}^{(L)}, \tilde{r}, x, z_{1:L}, y) = \sum_{l=0}^{L} \tilde{r}_l(x)\Big(\sum_{y'\neq y}[\mathbf{t}^{(l)}_{y'} + \frac{1}{K-1}]_+ + g(l, c_1, \dots, c_L, K)\Big).
\tag{258}
$$

It remains to be seen whether we can verify the consistency of a loss of this form.