# OpenReview forum: "Is the acquisition worth the cost? Surrogate losses for   Consistent Two-stage Classifiers"
_NeurIPS.cc/2025/Conference — NeurIPS 2025 spotlight_

### Official Review · Reviewer_ysUj · 2025-06-28

**Clarity:** 3
**Significance:** 3
**Originality:** 3
**Rating:** 5
**Confidence:** 4

**Summary:**

This paper addresses a very natural and increasingly common setting: we often have access to a cheap model that can handle most predictions, and a more expensive, better-informed model we'd rather only use when necessary. The authors formalize this as a two-stage classification setup, where the first classifier can either predict or defer to a second, stronger model at a fixed cost. They propose a new surrogate loss based on a hinge formulation that is proven to be consistent with the cost-aware 0–1–c loss. The theoretical results are backed by both synthetic experiments and an application involving routing math questions to either a small or large LLM, depending on difficulty.

**Questions:**

See the 'weaknesses' section.

**Ethical Concerns:**

["NO or VERY MINOR ethics concerns only"]

**Final Justification:**

The paper is technically solid, clearly written, and presents a well-motivated contribution with strong theoretical support. While the visualization section and a few presentation details can be improved, the planned revisions directly address these concerns. Overall, the work is of good quality and significance, and I remain supportive of its acceptance.

**Limitations:**

Yes

**Quality:**

4

**Strengths And Weaknesses:**

Strengths

The paper is technically solid and conceptually thorough. It’s also very well written. Not only are the motivations clear and compelling, but the notations, assumptions, and formulations are all introduced with careful justification and helpful intuition.
One of the nicest things about the paper is that it’s quite self-aware — it anticipates natural questions (like “why not use cross-entropy?”) and addresses them directly, both with formal analysis and experiments. The related work section is especially good at situating this work within the broader literature on deferral, dynamic inference, and loss consistency.
The theoretical contribution is well crafted. The surrogate loss is not only consistent, but the paper goes further and gives a negative result for the commonly used cross-entropy approach, showing that it cannot be Bayes-consistent with respect to the 0–1–c loss. This contrast makes the case for the proposed surrogate even stronger.

The LLM experiment, while not deeply benchmarked, is a smart choice that makes the whole problem setting feel concrete and relevant. Routing questions between small and large models based on difficulty is a task that many practitioners are actively thinking about.

Weaknesses

The visualization section could be improved. Figures 2, 3, and 4 require a fair amount of explanation to be fully understood, and they don’t clearly illustrate the practical difference between the hinge-based surrogate and the cross-entropy baseline. Visually comparing decision boundaries is tricky, and the figures don’t really speak for themselves. Since this is where the empirical case is made, it would help to make the differences more visually intuitive.

Related to that, it would be helpful to include the empirical 0–1–c risk of the cross-entropy model alongside the hinge-based one in Figure 4, even if the former is theoretically inconsistent. A quantitative comparison would help drive home the point about practical performance.
A few small presentation issues:
Line 122: the notation r: x → 0,1 is missing curly braces. Should be r: x → {0,1}.
Line 126: the definition of H is missing a closing brace.
The phrase “0–1 c loss” (e.g., in lines 198 and 302) can be hard to parse visually — it reads like “closs.” Consider reformatting it more clearly, perhaps using hyphens or math mode if applicable.

---

> ### Author Rebuttal · Authors · 2025-07-30
>
> Thank you for this thoughtful review, and for highlighting the clarity of our presentation and the contextualization of our work, as this was an important point for us.
> ## 1. The visualization section could be improved. Figures 2, 3, and 4 require a fair amount of explanation to be fully understood, and they don’t clearly illustrate the practical difference between the hinge-based surrogate and the cross-entropy baseline. Visually comparing decision boundaries is tricky, and the figures don’t really speak for themselves. Since this is where the empirical case is made, it would help to make the differences more visually intuitive.
>
> Yes, we agree that this could be made clearer. While we cannot provide new figures at this stage, we can describe the improvements we will make based on your comments. You are right that these points are central to our paper and should be presented as clearly and intuitively as possible.
>
> First, we will add a dashed line in Figure 1, labeled as the true decision boundary at a specific cost, to help readers better understand the optimal decision boundary and introduce it before Figures 2 and 3.
>
> Next, we will align and visually compare the true decision boundary, the boundary induced by our hinge loss, and the boundary produced by the cross-entropy loss. This will make it clearer that the cross-entropy loss fails to capture the optimal boundary, whereas our hinge loss closely matches it.
>
> ## 2.  Related to that, it would be helpful to include the empirical 0–1–c risk of the cross-entropy model alongside the hinge-based one in Figure 4, even if the former is theoretically inconsistent. A quantitative comparison would help drive home the point about practical performance. A few small presentation issues: Line 122: the notation r: x → 0,1 is missing curly braces. Should be r: x → {0,1}. Line 126: the definition of H is missing a closing brace. The phrase `0–1 c loss` (e.g., in lines 198 and 302) can be hard to parse visually — it reads like `closs.` Consider reformatting it more clearly, perhaps using hyphens or math mode if applicable.
>
>  Thank you for the suggestions and pointed out typos, we will integrated them. We will change the $0-1$ c loss to $01c$ loss.

---

> > ### Comment · Reviewer_ysUj · 2025-08-03
> > **Response**
> >
> > We thank the authors for their thoughtful rebuttal. The planned improvements to the visualization section, as well as the additional clarifications outlined, sound very promising and should make the paper even clearer and more accessible. We also appreciate how the responses to the other reviewers’ comments further address open questions and improve the overall presentation.
> >
> > Overall, we remain supportive of this paper’s publication and look forward to seeing the revised version.

---

### Official Review · Reviewer_nQwJ · 2025-07-03

**Clarity:** 3
**Significance:** 2
**Originality:** 3
**Rating:** 3
**Confidence:** 4

**Summary:**

This paper studies a two-stage classification setting where the first-stage model can defer to a second, more informed model at a cost. The authors introduce the $\ell_{01c}$ loss to model the trade-off between classification errors and deferral cost, and propose a hinge-based surrogate loss, $\ell^c_{\rm{hinge}}$, that is more tractable for training while ensuring consistency with the original objective. The approach enables joint training of both classifiers and is well-suited for applications leveraging foundation model features.

**Questions:**

1. How is the deferral cost to the second-stage (larger) model approximated  in the experiments?

2. Did the authors consider using smoother surrogates (e.g., softplus or  log-barrier variants) instead of the hinge loss to address optimization challenges with neural networks?

3. How does the proposed method empirically compare with models trained using cross-entropy-based loss functions?

4. How does the proposed methodology differ from, and compare to, the cascade framework presented in Jitkrittum et al., “When does confidence-based cascade deferral suffice?” NeurIPS 2023, which is applicable in the multi-stage setting? What advantages, if any, does the proposed method offer over this approach?

**Ethical Concerns:**

["NO or VERY MINOR ethics concerns only"]

**Final Justification:**

I have carefully considered the author's rebuttal, but my assessment remains a borderline reject.

My primary reservation is the practical applicability of the proposed hinge-based surrogate loss. The core issue stems from the non-differentiability of the hinge function, a property known to introduce significant computational inefficiencies during optimization, especially within deep learning frameworks. This is a well-established challenge that poses a major barrier to real-world adoption.

While I acknowledge the promising results in the additional experiments conducted under a simple setting, their limited scope does not sufficiently allay my concerns about generalizability and performance at scale. The fundamental question of whether this approach is viable for complex, real-world scenarios remains unanswered.

Therefore, I stand by my initial score. As also noted by another reviewer, the paper requires more extensive and rigorous empirical validation to substantiate the claims of its practical utility.

**Limitations:**

Yes.

**Paper Formatting Concerns:**

The formatting looks good to me.

**Quality:**

2

**Strengths And Weaknesses:**

Strengths:

1. The paper clearly defines a two-stage classification problem where a lightweight model can defer to a more costly model with access to richer features.

2. The paper introduces the $\ell_{01c}$ loss to explicitly encode trade-offs between prediction error and deferral cost, and derives a hinge-based surrogate loss ($\ell^c_{\rm{hinge}}$) that preserves consistency while being more tractable than the original discrete loss.

3. The paper offers a nuanced extension to the standard learning-to-defer paradigm by allowing richer second-stage inputs and unfreezing both classifiers.

Weaknesses:

1. The hinge-based surrogate is non-smooth, which may hinder optimization with neural networks. The paper does not discuss this practical issue or whether smooth approximations (e.g., log-barrier or softplus variants) were considered. The negative results reported in Section 4.1 further underscore this limitation.

2. The experiments do not include comparisons with more common baselines trained using cross-entropy loss, which would help contextualize the practical benefits of $\ell^c_{\rm{hinge}}$.

3. The results are restricted to the two-stage setting, whereas extending the approach to the more practical and potentially impactful multi-stage setting would be of greater interest.

---

> ### Author Rebuttal · Authors · 2025-07-30
>
> Thank you for engaging with our work and for highlighting the theoretical strengths of our paper. We have integrated the suggested baselines, added further discussion on the relevant topics and points you raised, and provided pointers toward the requested extension.
>
> ## 1.  The hinge-based surrogate is non-smooth, which may hinder optimization with neural networks. The paper does not discuss this practical issue or whether smooth approximations (e.g., log-barrier or softplus variants) were considered. The negative results reported in Section 4.1 further underscore this limitation.
>
> This is a valuable point. We will include additional discussion around the limitations of using the hinge loss, and suggest that smoother alternatives such as log-barrier or softplus [Dugas et al. (2001), Nock and Nielsen (2009), Zhang (2004)] variants could be explored as approximations if needed.
>
> We did provide experiments showing that the proposed loss can indeed be optimized in practice with neural networks. Both the LLM-based and the synthetic experiments used neural network models trained end-to-end with our loss, following architectures similar to those commonly used in real-world applications.
> [1] C. Dugas, Y. Bengio, P. Bélisle, C. Nadeau, and R. Garcia, “Incorporating second-order functional knowledge for better option pricing,” in Advances in Neural Information Processing Systems, vol. 13, 2001, pp. 472–478.
>
> [2] R. Nock and F. Nielsen, “A real log barrier primal-dual algorithm for large-scale nonlinear programming,” Optimization Methods and Software, vol. 24, no. 5, pp. 771–799, 2009.
>
> [3] T. Zhang, “Statistical behavior and consistency of classification methods based on convex risk minimization,” Annals of Statistics, vol. 32, no. 1, pp. 56–85, 2004.
>
>
>  ## 2. The experiments do not include comparisons with more common baselines trained using cross-entropy loss, which would help contextualize the practical benefits of lhinge
>
> In Section 4.1, we did present a cross-entropy–based version that directly targets our problem formulation, and we showed both theoretically and empirically that it fails to recover the correct decision boundary.
>
> If you are instead referring to the standard confidence-based approach (i.e., where a threshold is applied to the predicted probabilities of a separately trained model), we agree that this is a useful baseline for comparison, but it is also not consistent. We will integrate this method into our experiments as a baseline to better highlight the differences and limitations of such decoupled strategies in our setting. We've included preliminary results in our response to R1.3.
>
>
> ## 3. The results are restricted to the two-stage setting, whereas extending the approach to the more practical and potentially impactful multi-stage setting would be of greater interest.
>
> Other reviewers have also requested a similar extension. We will integrate into the paper more explicit guidance on how our results could be extended, by adding an Appendix. In our response to R2.5, we've explained how the result could be extended to defer to multi-models at the second stage. For the multi-stage case, which corresponds to a cascade setting, the extension would be similar but with some modifications.
>
> The decision function would become multiple decision functions, each making a decision at its respective stage and having access to increasingly more information. The classifiers would also have access to an increasing amount of data. This would be modeled as follows:
>
> \begin{align}
> \ell_{01c}(f(x,z_{1:L}), y) = \begin{cases}
> 1[f_1(x)\neq y] & r_0(x) = 0 \\\\
> 1[f_2(x,z_1)\neq y] + c_1 & r_1(x,x_1) = 1 \\\\
> \dots, \\\\
> 1[f_L(x,z_1, \dots, z_L)\neq y] + c_L & r_L(x,z_1, \dots, z_L) = L.
> \end{cases}
> \end{align}
>
>
> That said, we would like to emphasize two points: 1) Even with just two models, our framework already supports meaningful and practical applications. For example, scenarios involving edge devices and servers often naturally involve two models (one lightweight and one more powerful);  and  2) our work already represents a significant step forward, as it introduces the first consistency-based loss tailored for this setting. To the best of our knowledge, no prior work has addressed this challenge in a theoretically grounded way. This mimics how the learning to defer literature developed: early works first established the consistency of deferring to a single expert (Mozannar and Sontag [2020] , Verma et al. [2022]), and only later were extensions to multiple experts presented in Verma et al. [2022].
>
> ## 4. How is the deferral cost to the second-stage (larger) model approximated in the experiments?
>
>  The deferral cost is not approximated in our experiments; we present multiple experimental settings with various cost values. The reason for this choice is that the value of the cost parameter $c$ is inherently task-dependent. This cost can represent latency, energy consumption, or other computational resource metrics. It is therefore important to evaluate the method across a range of cost values to ensure that it performs consistently and does not fail in specific cost regimes.
>
> In the LLM experiment, we report results for several values of $c$, as shown on the $x$-axis of Figure 5, including both low and high values: $c = 0.01$, $0.1$, $0.3$.
>
>
>
> ## 5 Did the authors consider using smoother surrogates (e.g., softplus or log-barrier variants) instead of the hinge loss to address optimization challenges with neural networks?
>
>   No, as our focus was on providing a consistency surrogate loss. Using those approximations would break the consistency proof. We will include this as an approximation that could be used in practice if our hinge proposal is too troublesome, as we mentioned in our response to your first point.
>
> ## 6 How does the proposed method empirically compare with models trained using cross-entropy-based loss functions?
>
>  This result was provided in Section 4.1, where we explicitly showed that in a controlled environment, the cross-entropy version of the loss cannot recover the optimal boundary. If you are referring to other confidence-based methods, we have included preliminary results in our response to R1.3.
>
> ## 7 How does the proposed methodology differ from, and compare to, the cascade framework presented in Jitkrittum et al., ''When does confidence-based cascade deferral suffice?'' NeurIPS 2023, which is applicable in the multi-stage setting? What advantages, if any, does the proposed method offer over this approach?
>
> In short, the work by Jitkrittum et al. considers a fundamentally different setting and does not propose an approach for our setting.
>
> **Setting**: Jitkrittum et al. consider a cascade setting where the classifiers are **pre-trained and fixed**, and only the deferral rule is learned:
>     `we focus on a setting where the base models are pre-trained and fixed, and the goal is to train only the deferral rule.`
>
> In our setting, the classifiers are **not fixed** and are **jointly trained** with the deferral rule.
>
> **Approach:**  Jitkrittum et al. investigate scenarios where the confidence-based deferral rule is not optimal. To analyze this, they use post hoc rules designed to approximate the oracle deferral rule. However, these post hoc rules are not implementable in practice because they rely on access to information unavailable at decision time (such as the confidence of $f_2$). In our setting, this would correspond to having access to $z$ at no cost in order to make the decision to defer $r(x,z)$, which would render the problem nonsensical. Their results show that, indeed, in some settings, the confidence-based rule is suboptimal.
>
> Their work can be seen as a useful justification and motivation for the problem setting we address, while also revealing that in some cases joint training is not likely to provide substantial benefits.
>
> We directly contextualize the result from this work at line 153:
> `This result is similar to the solution for the decision rule given fixed classifiers first provided by Jitkrittum et al, which would read as $\max_y\eta_y(x) \geq \max_y\zeta_y(x,Z) -c$. However, our explicit modeling of the two-tiered information available to $f_1, r$ and $f_2$ provides a more practical and detailed solution, as it allows us to integrate the constraint that $r$ cannot fully access the information available to $f_2$. This modeling choice leads to a decision based on the \textbf{expected} future gain.`

---

> ### Comment · Reviewer_nQwJ · 2025-08-05
>
> Thank you to the authors for their responses. My primary remaining concern relates to the practical applicability of the proposed hinge-based surrogate. It is well known that hinge losses are less computationally efficient in practice—particularly in deep learning settings—due to their non-differentiability.
>
> While the experimental results presented are promising, it remains unclear whether these findings generalize to real-world scenarios, given the limited scope of evaluation. In particular, although you note that log-barrier and softplus variants break consistency guarantees, they are widely used in practice due to their smoothness and compatibility with gradient-based optimization. Why were these commonly used approximations not included as baselines in your experiments? If the hinge-based surrogate is indeed superior, such comparisons would help substantiate its practical value. In your experiments, does the hinge-based loss actually outperform these approximations?
>
> I view this comparison as both necessary and non-trivial, particularly given the well-documented challenges of optimizing hinge losses and the absence of these important baselines in your current experimental evaluation.
>
> As such, I will maintain my current evaluation unless such comparisons are provided.

---

> > ### Author Response · Authors · 2025-08-07
> >
> > We can provide additional experimental results to gain insight into how empirical performance would be impacted by using an approximation of our proposed hinge formulation.
> >
> > To that end, we use the softplus approximation and replace the max operator $max(0, x)$ in our loss with the softplus function:
> > $$ \text{softplus}(x) = \log(1 + e^{x}) $$
> > This gives us a new smooth approximation of our hinge loss:
> >
> > $$ \ell_{\text{softplus}}^{c}(\mathbf{t}, \mathbf{v}, \tilde{r}, x, z, y) = (1 - \tilde{r}(x)) \sum_{y' \neq y} \text{softplus}\left(\mathbf{t}_{y'} + \frac{1}{K - 1}\right)
> > $$
> >
> > $$+ \tilde{r}(x) \left(\sum_{y' \neq y} \text{softplus}\left(\mathbf{v}_{y'} + \frac{1}{K - 1}\right) + \frac{Kc}{K - 1} \right) $$
> > We present preliminary results on the synthetic task here. In the paper, we will provide additional visualizations to show that the loss does not converge to the optimal value, as our hinge version does.
> >
> >
> > #### Table 2. $\hat{R}_{01c}(f)$ with $K=5$
> >
> > | Baseline         | c = 0.03  | c = 0.05  | c = 0.07  |
> > |------------------|-----------|-----------|-----------|
> > | Softplus variant | 0.3771    | 0.3949    | 0.4092    |
> > | Hinge (ours)     | **0.3695** | **0.3777** | **0.3842** |
> >
> >
> > From our preliminary experiment on the synthetic task, which is perfectly solved by the hinge version, we can see that this simple approximation is not able to achieve results comparable to the hinge loss. These results are somewhat in line with the negative result we reported for cross-entropy in Section 4.1. As the structure of these losses (cross-entropy and softplus) is somewhat similar, we suspect that the same would hold for this softplus version; that it won’t be consistent. We then observe that for both of these non-consistent losses, the model is indeed not able to solve the problem.
> > We would like to emphasize that these negative results do not “further underscore this limitation.” On the contrary, they motivate our development of a consistent loss.

---

> > > ### Comment · Reviewer_nQwJ · 2025-08-07
> > >
> > > Thank you for the additional experiments using the softplus approximation. Could you also report the smooth approximation obtained by using the squared hinge loss, $\max(0, x)^2$, instead of the hinge loss, $\max(0, x)$? This would help clarify whether the inferior results are due to a poor approximation (which might be improved by using an alternative) or whether they reflect the superiority of the hinge loss itself (possibly due to its consistency, as shown by the authors).
> > >
> > > Additionally, could you please report the standard deviation to assess whether the results are statistically significant?

---

> > > > ### Author Response · Authors · 2025-08-07
> > > >
> > > > Here are the results for the softplus approximation, with standard deviations computed over 5 trials.
> > > >
> > > > #### Table 2. $\hat{R}_{01c}(f)$ with $K=5$
> > > >
> > > > | Baseline         | c = 0.03         | c = 0.05         | c = 0.07         |
> > > > |------------------|------------------|------------------|------------------|
> > > > | Softplus variant | 0.3771 ± 0.0010  | 0.3949 ± 0.0011  | 0.4092 ± 0.0014  |
> > > > | Hinge (ours)     | **0.3695 ± 0.0005** | **0.3777 ± 0.0005** | **0.3842 ± 0.0005** |
> > > >
> > > > We can include the  squared max variant in the final version as well.

---

> > > > > ### Author Response · Authors · 2025-08-08
> > > > >
> > > > > Our apologies, we thought that the deadline for the rebuttal was August 6th and missed that it was pushed to the 8th.
> > > > > Here are the results for the squared max variant as requested.
> > > > >
> > > > > The $\max^2$ baseline is obtained by swapping the softplus operator:
> > > > >
> > > > > $$ \ell^{c}(\mathbf{t}, \mathbf{v}, \tilde{r}, x, z, y) = (1 - \tilde{r}(x)) \sum_{y' \neq y} \text{max}(0,\left(\mathbf{t}_{y'} + \frac{1}{K - 1}\right))^2
> > > > > $$
> > > > >
> > > > > $$+ \tilde{r}(x) \left(\sum_{y' \neq y} \text{max}(0,\left(\mathbf{v}_{y'}+ \frac{1}{K - 1}\right))^2 + \frac{Kc}{K - 1} \right) $$
> > > > >
> > > > >
> > > > > We also include a different variant $max_{adjusted}^2$ with a squared penalty  $(\frac{Kc}{K - 1})^2  $ :
> > > > >
> > > > > $$ \ell^{c}(\mathbf{t}, \mathbf{v}, \tilde{r}, x, z, y) = (1 - \tilde{r}(x)) \sum_{y' \neq y} \text{max}(0,\left(\mathbf{t}_{y'} + \frac{1}{K - 1}\right))^2
> > > > > $$
> > > > >
> > > > > $$+ \tilde{r}(x) \left(\sum_{y' \neq y} \text{max}(0,\left(\mathbf{v}_{y'}+ \frac{1}{K - 1}\right))^2 + (\frac{Kc}{K - 1})^2 \right) $$
> > > > >
> > > > > We present preliminary results on the synthetic task here.
> > > > >
> > > > > #### Table 3. $\hat{R}_{01c}(f)$ with $K=5$
> > > > >
> > > > > | Baseline           | \(c = 0.03\)        | \(c = 0.05\)         | \(c = 0.07\)         |
> > > > > |--------------------|---------------------|----------------------|----------------------|
> > > > > | Softplus variant    | 0.3771 ± 0.0010     | 0.3949 ± 0.0011      | 0.4092 ± 0.0014      |
> > > > > | $max^2$           | 0.3948 ± 0.0030     | 0.3977 ± 0.0099      | 0.3949 ± 0.0022      |
> > > > > | $max_{adjusted}^2$ | 0.3948 ± 0.0031     | 0.3953 ± 0.0059      | 0.3951 ± 0.0064      |
> > > > > | Hinge (ours)        | **0.3695 ± 0.0005** | **0.3777 ± 0.0005**  | **0.3842 ± 0.0005**  |
> > > > >
> > > > > We can see that the three approximations of the hinge operator performed similarly.

---

### Official Review · Reviewer_C48H · 2025-07-03

**Clarity:** 3
**Significance:** 2
**Originality:** 2
**Rating:** 3
**Confidence:** 4

**Summary:**

The problem this paper tries to solve is as follows: we have a classifier f1 that uses input x and a classifier f2 that uses input z, however the classifier f2 incurs a cost c>0 every time it is used, how can we learn the pair (f1,f2) to minimize the misclassification loss taking into account the cost c. This translates to a loss function l_{01c} which is non-differentiable, this paper aims to create a surrogate loss that is consistent with this loss and easy to optimize. They first outline the optimal bayes solution to this problem. They then propose a hinge based surrogate loss function that is consistent with the l_{01c} loss function. Moreover, they prove that a cross-entropy like surrogate loss would not be consistent. They evaluate their new surrogate on synthetic experiments and an real experiment using features from 7b (x) and 70b (z) llama models on a multiple choice math dataset.

**Questions:**

- can you add baselines to the experiments? For instance simple confidence based thresholding without training the models.

- Is it possible to conduct more comprehensive experiments in settings that mimic where this loss might be useful for? For instance an NLP experiment with small bert models (f1) vs llama models (f2) that are fine-tuned with Llora instead of taking the embeddings only? Other experiments are welcome.

- Can you comment on how to extend to defer to multiple models each with different costs c_i ?

**Ethical Concerns:**

["NO or VERY MINOR ethics concerns only"]

**Final Justification:**

I have read the author's rebuttal. I remain on my original score of borderline reject. The authors note that the main contribution is theoretical one, however, as a researcher in this area I need to see  some convincing empirical evidence that this new loss function is worth it compared to baselines before I invest time implementing it for my own work. I don't think the loss function on it's own is differentiated in terms of theory compared to prior work in this area to stand on it's own without empirical validation. I think this papers fails to provide that empirical evidence, many other papers in this area do have theoretic contributions supplemented with empirical validation.

**Limitations:**

yes

**Quality:**

2

**Strengths And Weaknesses:**

Clarity: The paper is easy to read and clear.

Quality: The theoretical results of this paper are sound. The empirical results however fall short of providing sufficient proof that the method works better than simple baselines. The LLM experiments in Section 5.2 is quite limited: it lacks baselines, it evaluates a single setup on a single dataset and is with a small sample size. Given the limited empirical evidence, it is hard to judge whether this loss function is effective.

Originality: The loss function derived is novel. The technique to derive the loss function extends existing working on learning to defer to derive this hinge based loss.

Significance: The problem this paper is trying to solve has many real world examples. For instance, routing between different LLM models or model providers is likely to be one real world application. However, the empirical experiments do not provide enough evidence to use this loss function in practice.

---

> ### Author Rebuttal · Authors · 2025-07-30
>
> Thank you for engaging with our work and for your thoughtful feedback.
> Most of the concerns raised relate to the empirical results. While we appreciate these insights and have done our best to incorporate the suggestions where possible, we would like to emphasize that the primary contribution of this paper is theoretical. The empirical results are included mainly to provide insight into the nature of the theoretical findings, rather than to offer exhaustive experimental proof of the practical superiority of the proposed loss function. Even without extensive empirical comparisons, specifying a consistent surrogate loss function is an important step that can lay the groundwork for other researchers to design training schemes and loss adaptations or approximations that improve practical performance. There are similar works in this line of research that are primarily theoretical and also provide relatively limited empirical studies (e.g., Long and Servedio, 2013; Mao et al., 2023; Cao et al., 2022). Our paper follows a similar approach, aiming to advance theoretical understanding while including illustrative experiments to support key insights.
>
> We believe this type of theoretical paper is as valuable as more empirical works and constitutes a meaningful contribution to the NeurIPS conference.
> <!-- Most of the criticisms below pertain to the empirical results. While we appreciate the insights, and have done our best to accommodate the suggestions, and believe that many of your points are very valuable indicators for future work, we would like to emphasize that the contribution of this paper is primarily theoretical. Empirical results are included to provide insights into the nature of the theoretical results, rather than to demonstrate thorough experimental proof that the proposed loss function is practically superior. Even without an exhaustive empirical comparison, the specification of a consistent surrogate loss function is an important step that can provide a foundation for other researchers to design training schemes and loss adaptations/approximations that yield improved practical performance.
>
> In numerous settings in the machine learning literature, some papers make primarily theoretical contributions while others are primarily experimental. For example, in the learning to defer setting, there are multiple papers that provide theoretical results concerning consistency without including substantial experimental studies (e.g., Long and Servedio, 2013, Mao et al., 2023, and Cao et al., 2022).
> In most of these papers, the experimental results are considerably more limited than what we report. The expectation that a single paper do both is somewhat unreasonable - the vast majority of papers at NeurIPS would be rejected according to these criteria, as very few provide both extensive novel theoretical contributions and expansive empirical investigations.  -->
>
> ## 1.  The theoretical results of this paper are sound. The empirical results however fall short of providing sufficient proof that the method works better than simple baselines. The LLM experiments in Section 5.2 is quite limited: it lacks baselines, it evaluates a single setup on a single dataset and is with a small sample size. Given the limited empirical evidence, it is hard to judge whether this loss function is effective.
>
>  We acknowledge that the scope of our empirical results is limited, and we will include an additional baseline as suggested (see preliminary results in our response R3.1). That said, we would like to emphasize that the main contribution of this paper is theoretical. The experiment involving LLMs is intended to demonstrate that the proposed loss can be applied in practice and leads to an intuitive separation of inputs.  We recognize that this is not a comprehensive empirical study comparing our method to a wide range of baselines.
>     It certainly is of interest to perform an experimental study of our proposed loss function, exploring practical scenarios, but we believe that it is more appropriately left for future work.
>
> The goal of our included experiments is to provide insight into the behavior of the learned functions $f_1, f_2, r$ when trained with our surrogate loss. One of the insights provided by the experiment we include can be seen in our results in Figure 5, where we observe a ''specialization'' of the classifiers. This behavior cannot arise for example  when using the simple baseline of confidence-based thresholding, since in that setting the models are trained independently and lack any mechanism for joint specialization/coordination.
>
>
> ## 2. Significance: The problem this paper is trying to solve has many real world examples. For instance, routing between different LLM models or model providers is likely to be one real world application. However, the empirical experiments do not provide enough evidence to use this loss function in practice.
>   See our response 1.
>
> ## 3 can you add baselines to the experiments? For instance simple confidence based thresholding without training the models.
>
> Yes, thank you for this suggestion. We can add this as a baseline for the LLM experiment. We have included preliminary results of the suggested baselines on the synthetic experiment in our response to R1.3
>
> ## 4 Is it possible to conduct more comprehensive experiments in settings that mimic where this loss might be useful for? For instance an NLP experiment with small bert models (f1) vs llama models (f2) that are fine-tuned with Llora instead of taking the embeddings only? Other experiments are welcome.
>
> See response to 2.1. The outlined experiment is indeed an excellent suggestion, but we don't think it is practical to embark on a completely new experiment of this relatively extensive nature during the rebuttal period.
>
> ## 5 Can you comment on how to extend to defer to multiple models each with different costs $c_i$ ?
>
> We believe that this result can indeed be obtained, but it would require a non-trivial amount of additional work to prove formally. It represents a valuable direction for future research, and we will add more details in the paper to outline how our results could  be extended in that direction. We will add an Appendix containing the following details:
>
> To extend our result to $L+1$ classifiers, where there are still two stages, but there are $L$ classifiers to choose from at the second stage, we would need to introduce additional random variables $Z_1, \dots, Z_L$, and the loss in Eqn.2 would need to be generalized to multiple classifiers and costs:
>
> \begin{align}
> \ell_{01c}(f(x,z_{1:L}), y) = \begin{cases}
> 1[f_1(x)\neq y] & r(x) = 0 \\\\
> 1[f_2(x,z_1)\neq y] + c_1 & r(x) = 1 \\\\
> \dots, \\\\
> 1[f_L(x,z_L)\neq y] + c_L & r(x) = L.
> \end{cases}
> \end{align}
>
> The corresponding solution for the decision boundaries (Eqn. 9) would become more complex. Instead of comparing the maximum posterior probability $\eta(x)$ to a single expected future gain minus cost, the comparison would now need to be made against **the best** potential expected future gain for each classifier:
>
> \begin{align}
> r^*(x) = \begin{cases} 0 &\text{ if } 1[\max_y \eta_y(x) > \max_{l \in {1,L}} (E_{p(Z_l|x)}[ \max_y p(y|x,Z_l)] - c_l ) ] \\\\
> 1 &\text{ if } 1 = \max_{l \in {1,L}} (E_{p(Z_l|x)}[ \max_y p(y|x,Z_l)] - c_l )  \\\\
> \dots \\\\
> L &\text{ if } L = \max_{l \in {1,L}} (E_{p(Z_l|x)}[ \max_y p(y|x,Z_l)] - c_l )
> \end{cases}
> \end{align}
>
> The term $\max_{l \in {1,L}} \Big(\mathbb{E}_{p(Z_l|x)}[ \max_y p(y|x,Z_l)] - c_l \Big)$ returns the index of the best model that can be used at the second stage.
>
> Then, we could propose a multi-classifier surrogate loss (replacing Eqn. 11) by using a soft decision function that is now a multiclass probability vector $\tilde{r}(x) \in [0,1]^{L+1}$ with $\sum^L_{l=0} \tilde{r}^l(x) = 1$. We would also need to introduce $L$ new learnable vectors $\mathbf{t}^{(0)} ,  \dots, \mathbf{t}^{(L)}$, and an index-dependent cost term $\frac{K}{K-1}g(l, c_1, \dots, c_L, K)$, where $g(l, c_1, \dots, c_L, K)$ is some linear function of the costs that would need to be derived and obtained from the proof. The hinge surrogate loss could (potentially) have the following form:
>
> \begin{align}
> \ell_{hinge}^{c}( \mathbf{t}^{(0)}, \dots, \mathbf{t}^{(L)}, \tilde{r}, x, z_1, \dots, z_L, y)  = \sum^L_{l=0} \tilde{r}^l(x) ( \sum_{y' \neq y} [t^{(l)}{y'} + \frac{1}{K{-}1} ]_+ + g(l, c_1, \dots, c_L, K) ).
> \end{align}
> We would then follow similar steps to those in the paper to verify its consistency.
>
> These formulations would differ if the multiple models were used in a cascade of L+1 stages, so that we have multiple choices to make at different levels in the cascade, with decisions being made based on cumulative information. In such a setting, there would be a set of soft decision functions, with each $\tilde{r}_i(\cdot) \in [0,1]$. Insights into this extension was requested by R3.3, so we provide the decision function there.

---

### Official Review · Reviewer_uhgW · 2025-07-03

**Clarity:** 3
**Significance:** 3
**Originality:** 3
**Rating:** 5
**Confidence:** 3

**Summary:**

This paper explores a two-stage classification framework in which the first classifier, operating on a basic feature set, can either issue a prediction or defer the decision to a second, more informed classifier that has access to additional features, at an associated cost. To enable effective joint training of both classifiers, the authors propose a hinge-based surrogate loss function that approximates the original cost-sensitive objective while remaining amenable to optimization.

**Questions:**

1. The loss function presented in Equation (11) could benefit from a clearer explanation. Adding more intuition or context around its components would help improve the clarity and readability of the paper.
2. The paper introduces a cost c for deferring to the second-stage classifier. Could the authors clarify whether this represents computational inference cost, monetary cost, or some other type of penalty?
3. While the paper references related methods, such as learning to defer and learning to abstain, it does not include empirical comparisons to these approaches. Including such comparisons would significantly strengthen the evaluation and highlight the advantages or trade-offs of the proposed method.

**Ethical Concerns:**

["NO or VERY MINOR ethics concerns only"]

**Final Justification:**

The authors addressed the main concerns raised in the initial review. They added relevant baselines, clarified the proposed loss function, and acknowledged fairness implications and multi-stage extensions. While evaluation on LLMs lacks ground truth, this is a known challenge and does not significantly weaken the contribution.

**Limitations:**

The authors acknowledge the main limitation that the work is restricted to a two stage setting and suggest extending to multi stage systems as a future direction. While the societal impact is not explicitly discussed, I believe the work does not pose immediate risks and focuses on improving efficiency and consistency in model deployment. Adding a brief discussion on how cost sensitive deferral strategies might affect fairness or access (e.g., systematically deferring difficult cases) would strengthen this section.

**Paper Formatting Concerns:**

The acronyms are often inconsistent, could use a deeper look. The figures could be better represented. For instance, the legend can be at the bottom instead of inside the fig. Providing pseudocode where needed can help understand the flow better.

**Quality:**

3

**Strengths And Weaknesses:**

1. The paper discusses a practically important of jointly training a deferral mechanism between classifiers with different costs and capabilities. It introduces a hinge-based surrogate loss and rigorously proves its Bayes consistency with respect to the cost-aware 0-1 loss.
2. Unlike previous works on learning to defer or reject, this approach handles joint training of multiple classifiers, rather than assuming access to fixed experts. The proof that no cross-entropy variant can be consistent for this loss is a useful result.
3. The paper includes both synthetic and realistic LLM-based experiments that demonstrate the practical relevance of the proposed approach.
4.  The framework and analysis are restricted to two classifiers; many real-world adaptive systems involve more than two stages or models.
5.  The synthetic experiment is clean but limited in scope, and the LLM experiment has no ground-truth optimal decision for deeper evaluation.
6. There’s a lack of comprehensive comparisons to other recent surrogate-based or dynamic routing methods (e.g., soft deferral, confidence thresholding).

---

> ### Author Rebuttal · Authors · 2025-07-30
>
> Thank you for your positive review and for highlighting the strong theoretical foundation of our work. We found your feedback valuable and have addressed the points you raised in our response.
>
> ## 1. The framework and analysis are restricted to two classifiers; many real-world adaptive systems involve more than two stages or models.
>
>  We acknowledge that extending our results to more than two classifiers is a very interesting and worthwhile direction. While developing a full set of new theoretical results would require substantial work and is better suited for a follow-up paper, we will include in the appendix a more thorough discussion and concrete pointers on how one might extend our theoretical to the multi-stage classification setting. (See our response to R2.5 for more details.)
>
> ## 2. The synthetic experiment is clean but limited in scope, and the LLM experiment has no ground-truth optimal decision for deeper evaluation.
>
> Unfortunately, for any experiment involving real data, it is not possible to derive the ground-truth optimal decision boundary. To approximate this optimal decision boundary within our framework, we would need access to the true posterior distributions $p(Y|x), p(Y|x,z)$. However, these quantities are not available unless we assume that our learned model is perfectly calibrated. If we do make this assumption, the evaluation becomes circular (effectively assuming the model is correct in order to evaluate that same model).
> As a result, the accuracy and deferral rates we report are the only reliable evaluation metrics in this setting.
>
>
> ## 3. There’s a lack of comprehensive comparisons to other recent surrogate-based or dynamic routing methods (e.g., soft deferral, confidence thresholding).
>
> Confidence thresholding (CT) and soft deferral are both methods designed for the **fixed classifier** setting (not the joint learning setting we target). However, we agree that including them in the experiments would strengthen our presentation. We will therefore include both a confidence-based thresholding method and a soft deferral method as baselines.
>
> We describe below the additional baselines that will be added to the paper.
> Let $acc(f)$ denote the empirical accuracy of a model $f$ evaluated on a validation set.
>
> - **CT-c:** We pretrain both $f_1$ and $f_2$ using cross-entropy (CE). For a given $c$, we set the threshold $\tau = acc(f_2) - c$, so the deferral rule is defined as $r(x) = 1[\max_y \eta_y(x) < \tau]$.
>
> - **CT:** We pretrain both $f_1$ and $f_2$ using cross-entropy (CE), and search for the optimal $\tau$ using a validation set.
>
> - **Soft deferral:** We pretrain both $f_1$ and $f_2$ using cross-entropy (CE). The deferral rule is sampled per example using $p = 1 - \max_y \eta_y(x)$ as the Bernoulli parameter:
>   $r(x) \sim \text{Bernoulli}(p = 1 - \max_y \eta_y(x))$.
>
>  - **L2D:** We pretrain $f_2$ using cross-entropy (CE), and set it as an expert for the method [Mozannar and Sontag, 2020].
>
> ---
>
> We present preliminary results on the synthetic experiment by reporting the empirical risk $\hat{R}_{01c}(f)$. Since the rebuttal format limits us to tabular display, we report only the empirical risk of the four baselines. These results will be added to Figure 4, and Figure 6 will show corresponding results for the LLM experiment.
>
> #### Table 1. $\hat{R}_{01c}(f)$ with $K=5$
>
> | Baseline      | c=0.03     | c=0.05     | c=0.07     |
> |---------------|------------|------------|------------|
> | CT-c          | 0.3701     | 0.3842     | 0.4031     |
> | CT            | 0.3700     | 0.3784     | 0.3853     |
> | Soft deferral | 0.3700     | 0.3789     | 0.3855     |
> | L2D |  0.3700  |  0.3788   |   0.3857   |
> | Hinge (ours)         | **0.3695** | **0.3777** | **0.3842** |
>
> ---
>
> As shown in the table, across different values of $c$, the proposed surrogate loss consistently achieves the lowest empirical risk. Notably, it even outperforms the confidence-threshold baseline, which uses a validation set to tune its threshold. Although the difference is small, the goal is to supplement and illustrate the theoretical results.
>
>
> ## 4. The loss function presented in Equation (11) could benefit
> from a clearer explanation. Adding more intuition or context around its components would help improve the clarity and readability of the paper.
>
>   Thank you for that suggestion. We will add the following contextualization text in line 175 :  `The loss is composed of a sum of two terms: the first trains the first classifier, and the second trains the second classifier. The balance or weight assigned to each term on a per-sample basis is intuitively controlled by the learned soft decision $\tilde{r}$. If $\tilde{r}(x)$ indicates that a sample should be inferred by $f_1$, then $f_1$ will receive more weight during training at that point. In the second term, corresponding to $f_2$, we include an additional fixed term $\frac{Kc}{K-1}$ that encodes the penalty of using the second classifier. This encourages $\tilde{r}(\cdot)$ to favor the first term unless the benefit of using $f_2$ outweighs the cost. `
>
> ## 5. The paper introduces a cost c for deferring to the second-stage classifier. Could the authors clarify whether this represents computational inference cost, monetary cost, or some other type of penalty?
>
> We don't assume any particular form of cost, as it can represent any of the examples you listed. This cost is highly task-dependent and should be determined by the practitioner. This is illustrated in the example provided on line 111: `For instance, in recommendation systems, different types of user data queries can vary significantly in terms of latency and infrastructure expense.`
> Ultimately, the user should decide how much an error ($f(x) \neq y$) costs to the system compared to the cost of using a more powerful architecture (or acquiring/using more information), as modeled by $f_2$. As an example, consider a fraud detection system. The first classifier can respond in 10ms. The second classifier requires 150 ms (e.g. round-trip time to a more powerful remote classifier). A practitioner may decide that this additional 140ms latency penalty is only worthwhile for a given transaction if the expected improvement in accuracy exceeds 10 percent, motivating a setting of c=0.1.
>
> ## 6. While the paper references related methods, such as learning to defer and learning to abstain, it does not include empirical comparisons to these approaches. Including such comparisons would significantly strengthen the evaluation and highlight the advantages or trade-offs of the proposed method.
>
> Thank you for this suggestion. We will include additional baselines, as suggested by other reviewers as well. In particular, we will compare against learning-to-defer methods by pretraining the $f_2$ function and then treating it as an expert. (See our response to your point 3).
>
>  ## 7. The authors acknowledge the main limitation that the work is restricted to a two stage setting and suggest extending to multi stage systems as a future direction. While the societal impact is not explicitly discussed, I believe the work does not pose immediate risks and focuses on improving efficiency and consistency in model deployment. Adding a brief discussion on how cost sensitive deferral strategies might affect fairness or access (e.g., systematically deferring difficult cases) would strengthen this section.
>
>  Thank you for this suggestion, this is a fair point and we will include additional text to address the societal impact of how cost sensitive deferral strategies can affect fairness or access. In particular we will add a social impact section with the following text:
>
>      Although this is a theoretical paper and we believe poses minimal direct societal impact, the broader problem of cost-sensitive deferral systems may raise concerns related to fairness and access. In such systems, the model determines whether a query is ``simple'' and can be handled by a smaller model, or ``difficult'' and should be deferred to a more powerful model, which may involve higher computational cost or latency. This can introduce bias in how different users' queries are treated. For instance, if a particular user or group systematically submits queries that the system deems ``hard'', they may consistently experience greater latency, potentially leading to unfair treatment or limited access. Additionally, this introduces new potential pathways for bias to enter the system, as the deferral rule itself can be biased. This could further exacerbate disparities in user experience and overall system fairness.

---

> > ### Comment · Reviewer_uhgW · 2025-08-06
> >
> > Thank you for the response, I am maintaining my score. The authors added relevant baselines, clarified the loss function, and committed to expanding the fairness discussion and multi-stage extensions, which addresses the points I raised. While some limitations remain (like ground truth evaluation in LLMs), I continue to believe this work is a valuable contribution.

---

### Note · Authors · 2025-08-13

We thank the reviewers for their feedback and engagement during the rebuttal process.

All agreed our work is novel, the problem well-motivated and clearly defined, and our presentation clear. Importantly, all confirmed that our theoretical section is sound and well-supported. Our key result: the first consistent surrogate loss for jointly optimizing a cost-aware framing with a decision function, was recognized as strong, well-explained, and technically rigorous.
During the rebuttal, we addressed concerns about the limited empirical section by substantially strengthening our experiments. We added six new baselines: three differentiable approximations of our hinge loss and three related baselines (Learning to Defer, confidence thresholding, and soft deferral). These additions directly address the major criticism and provide a stronger comparison, further illustrating the advantages of our joint optimization approach. Moreover, we provided detailed explanations on how our results could be extended to different multi-classifier scenarios requested by reviewers.

We believe these additions and clarifications fully address the reviewers’ concerns and strengthen our work.

---

### Decision · Program_Chairs · 2025-09-17

**Decision:**

Accept (spotlight)

**Comment:**

This paper addresses a natural and increasingly common setting: we often have access to a cheap model that can handle most predictions, and a more expensive, better-informed model we'd rather only use when necessary. The authors formalize this as a two-stage classification setup, where the first classifier can either predict or defer to a second, stronger model at a fixed cost. They propose a new surrogate loss based on a hinge formulation that is proven to be consistent with the cost-aware 0–1–c loss. The theoretical results are backed by both synthetic experiments and an application involving routing math questions to either a small or large LLM, depending on difficulty.

The paper is theoretical in its approach but aims to address a practical problem. As such, the criticism of the experimental results and the potential hurdles in trying to implement its suggested paradigm are more relevant than in purely theoretical papers. However, overall, I think that the novelty of formulating the problem in a reality-relevant way and providing clear theoretical analysis outweighs those concerns.